# Swordtail fish hybrids reveal that genome evolution is surprisingly predictable after initial hybridization

Quinn K. Langdon[1,2¤]*, Jeffrey S. Groh[3], Stepfanie M. Aguillon[1,2,4], Daniel L. Powell[1,2,5], Theresa Gunn[1,2], Cheyenne Payne[1,2], John J. Baczenas[1], Alex Donny[1,2], Tristram O. Dodge[1,2], Kang Du[6], Manfred Schartl[6,7], Oscar Ríos-Cárdenas[8], Carla Gutiérrez-Rodríguez[8], Molly Morris[9], Molly Schumer[1,2,10]*

1 Department of Biology, Stanford University, Stanford, California, United States of America, 2 Centro de Investigaciones Científicas de las Huastecas "Aguazarca", A.C., Calnali, Hidalgo, Mexico, 3 Center for Population Biology and Department of Evolution and Ecology, University of California, Davis, Davis, California, United States of America, 4 Department of Ecology and Evolutionary Biology, University of California, Los Angeles, Los Angeles, United States of America, 5 Department of Biological Sciences, Louisiana State University, Baton Rouge, Louisiana, United States of America, 6 Xiphophorus Genetic Stock Center, Texas State University San Marcos, San Marcos, United States of America, 7 Developmental Biochemistry, Biocenter, University of Würzburg, Würzburg, Germany, 8 Red de Biología Evolutiva, Instituto de Ecología, A.C., Xalapa, Veracruz, Mexico, 9 Department of Biological Sciences, Ohio University, Athens, Ohio, United States of America, 10 Freeman Hrabowski Fellow, Howard Hughes Medical Institute, Stanford, California, United States of America

¤ Current address: Gladstone Institute of Virology, Gladstone Institutes, San Francisco, California, United States of America

* qlangdon@gmail.com (QKL); schumer@stanford.edu (MS)

**Data Availability Statement:** Ancestry probability files, recombination maps, wavelet analysis, and genome annotation files are deposited on Dryad (doi: 10.5061/dryad.qnk98sfq1). Custom code is

## Abstract

Over the past 2 decades, biologists have come to appreciate that hybridization, or genetic exchange between distinct lineages, is remarkably common—not just in particular lineages but in taxonomic groups across the tree of life. As a result, the genomes of many modern species harbor regions inherited from related species. This observation has raised fundamental questions about the degree to which the genomic outcomes of hybridization are repeatable and the degree to which natural selection drives such repeatability. However, a lack of appropriate systems to answer these questions has limited empirical progress in this area. Here, we leverage independently formed hybrid populations between the swordtail fish *Xiphophorus birchmanni* and *X. cortezi* to address this fundamental question. We find that local ancestry in one hybrid population is remarkably predictive of local ancestry in another, demographically independent hybrid population. Applying newly developed methods, we can attribute much of this repeatability to strong selection in the earliest generations after initial hybridization. We complement these analyses with time-series data that demonstrates that ancestry at regions under selection has remained stable over the past approximately 40 generations of evolution. Finally, we compare our results to the well-studied *X. birchmanni* × *X. malinche* hybrid populations and conclude that deeper evolutionary divergence has resulted in stronger selection and higher repeatability in patterns of local ancestry in hybrids between *X. birchmanni* and *X. cortezi*.

available on Zenodo (doi: 10.5281/zenodo. 12533281). Genome assemblies and raw data are deposited on the NCBI SRA (Bioproject PRJNA1106506).

**Funding:** This work was supported by a Sloan Fellowship to MS and NIH grant 1R35GM133774 to MS, NSF IBN grant 9983561 and Ohio University Research Incentive to MRM, a Stanford Tinker Award to MS and CRG, NSF Postdoctoral Research Fellowship in Biology (2010950) to QKL, and Stanford Science Fellowship to SMA. JG was supported by NIH grant R35 GM136290 to G. Coop. MS is a Freeman Hrabowski Scholar at Howard Hughes Medical Institute. The funders had no role in study design, data collection and analysis, decision to publish, or preparation of the manuscript.

**Competing interests:** The authors have declared that no competing interests exist.

**Abbreviations:** ABC, Approximate Bayesian Computation; LD, linkage disequilibrium; MAP, maximum a posteriori; TE, transposable element.

## Introduction

Hybridization has made substantial contributions to the genomes of species across the tree of life. Dozens of studies over the past 2 decades have documented pervasive genetic exchange between closely related species within all major eukaryotic groups [1–8]. Hybridization has even played an important role in the evolutionary history of our own species [9–11] and that of our close relatives [12,13]. Because we now know that genetic exchange between species is pervasive, unraveling the genetic and evolutionary impacts of hybridization is a fundamental part of understanding the genomes of modern species. The study of hybridization does not simply inform our understanding of evolutionary history, but is crucial for characterizing the genetic and evolutionary causes of traits, adaptations, and diseases in many species groups [14–18]. Moreover, characterizing the genomic consequences of hybridization promises to directly inform our understanding of the genetic changes that lead to divergence between species.

Modern genomic approaches to studying hybridization are often based on inference of local ancestry, or the ancestral source population from which a haplotype was derived, using genomic similarity to contemporary reference populations. With these approaches, researchers have moved from documenting evidence of hybridization in the genome as a whole to characterizing patterns of local variation in ancestry along the genome. Research into the history of genetic exchange between modern humans and our extinct relatives, the Neanderthals and Denisovans, was among the first to rigorously evaluate where in the genome ancestry from other lineages has been retained and where it has been lost [10,19–22]. This question has since been tackled in several species groups, including swordtail fish [23,24], *Saccharomyces* yeast [25], monkeyflowers [3,26,27], *Drosophila* [28], *Formica* ants [29], honey bees [5], *Heliconius* butterflies [30,31], and baboons [32]. Although the organisms in which these questions have been studied are diverse, some unifying observations have emerged from this work, hinting at shared principles that impact the predictability of genome evolution after hybridization. First, in most species studied to date, haplotypes that originate from the "minor" parent species, or the species from which hybrids derive less of their genome, are inferred to be on average deleterious (due to a number of possible mechanisms of selection; see below, [20,33,34]). Second, genome architecture seems to play a repeatable role in the purging of minor parent ancestry following hybridization. Researchers have consistently found that regions of the genome with low rates of recombination retain lower levels of minor parent ancestry, presumably because long introgressed haplotypes are more likely to contain multiple linked deleterious variants and thus be purged by selection more rapidly [3,29,30,32,35,36]. Theoretical studies have demonstrated that these dynamics are expected from first principles [37]. Similarly, researchers have found that regions of the genome especially dense in functional basepairs, including coding, conserved, and enhancer regions, are often depleted in minor parent ancestry [10,20,22,33,35] (but see [38] for discussion of the challenges of these analyses). Together, these observations point to a shared role of genome organization in the patterning of ancestry in the genome after hybridization.

These patterns highlight shared factors that drive genome evolution after hybridization across diverse taxa. However, it is still unclear whether selection drives repeatable patterns of local ancestry in replicated hybridization events between the same species, after accounting for these factors. From first principles, we might expect more repeatability in local ancestry across replicated hybrid populations in scenarios when more loci are under selection in hybrids [20], when sites under selection are shared, and when selection is strong relative to genetic drift [35,37]. The specific mechanisms of selection on hybrids are also likely to play an important role in the degree to which we expect repeatability in local ancestry in replicate hybrid

populations. In cases where selection on hybrids is largely driven by selection on negative epistatic interactions between substitutions that have arisen in the parental species' genomes (so-called "Dobzhansky–Muller" hybrid incompatibilities; but see [39]) or directional selection against one ancestry state (e.g., due to an excess of deleterious mutations that have accumulated along that lineage; [20,29,34,37]), we might predict that selection will drive repeatable ancestry patterns around selected sites. Moreover, theory and available empirical data predicts that the number of hybrid incompatibilities will increase nonlinearly with divergence between lineages [40–45], such that hybrid incompatibilities may play a larger role in the genome evolution of hybrids formed between distant relatives. By contrast, in species where selection against hybrids is largely dependent on the ecological environment [46,47], we might predict that selection will drive distinct patterns of local ancestry in distinct environments. The demographic history of the hybrid population itself is also crucial for interpreting signals of repeatability, since variables such as the time since admixture determine the scale of ancestry variation along a chromosome and the accumulated effects of genetic drift. However, temporally localized effects of selection can leave lasting impacts on ancestry variation, suggesting that ancestry patterns studied even long after admixture can be informative about the early stages of selection on hybrids [38,48].

Beyond the diverse biological factors at play, progress in understanding the repeatability of replicate hybridization events has been limited by the fact that only a handful of empirical studies have tackled this question. This is in part due to a lack of appropriate systems to test these questions (e.g., those with truly independent hybridization events) and in part due to the difficulty of excluding technical factors impacting the accuracy of ancestry inference that could be misinterpreted as biological signal. We focus our discussion here on studies that directly infer local ancestry states along the genome because of their precision and improved ability to distinguish hybridization from other biological processes (e.g., incomplete lineage sorting, background selection; [14,49,50]). However, we note that other approaches have provided important insights into the repeatability of genetic and phenotypic evolution after hybridization [3,31,44,51–54].

Some of the earliest studies to address questions about repeatability of local ancestry patterns asked whether there were shared deserts of archaic ancestry (i.e., Neanderthal and Denisovan ancestry) in the human genome [10,19]. These studies identified concordant patterns in the locations of deserts of archaic ancestry and the types of regions that harbor higher levels of archaic ancestry [10,19]. However, interpretation of these results is complicated by the challenges of distinguishing between Neanderthal and Denisovan ancestry [55], and other technical considerations [21]. Outside of hominins, 3 studies have explicitly inferred local ancestry and used it to evaluate the repeatability of genome evolution in replicated hybridization events. In *Drosophila*, Matute and colleagues showed that experimental hybrid populations generated between *Drosophila* species showed repeatable patterns of purging of minor parent ancestry [56]. In hybrid swarms generated between these species, ancestry from one parental species was consistently purged, and the regions where minor parent ancestry tracts were retained showed some level of repeatability in replicate populations. In replicate natural populations of hybrid ants that have evolved independently for tens of generations, researchers found remarkably high repeatability in local ancestry patterns across 3 hybrid populations, driven in part by selection against deleterious load inherited from one of the parental species [29]. Past work from our group asked about repeatability in patterns of minor parent ancestry in naturally occurring *Xiphophorus birchmanni* × *X. malinche* populations that formed independently in different river systems [57]. We found moderate predictability in local ancestry patterns between replicate *X. birchmanni* × *X. malinche* populations [17,58]. We also compared patterns of local ancestry between *X. birchmanni* × *X. malinche* hybrid populations to a hybrid

population of a different type, formed between *X. birchmanni* and its more distant relative, *X. cortezi* [57], and identified weak but significant correlations in local ancestry between hybrid population types.

Here, we identify a new independently formed hybrid population between *X. birchmanni* and *X. cortezi* (Fig 1A), allowing us to ask questions about how repeatability of genome evolution scales with increasing genetic divergence between hybridizing species. We observe an extraordinary level of repeatability in local ancestry patterns across independently formed *X. birchmanni* × *X. cortezi* hybrid populations, consistent with remarkably strong selection on hybrids. We find that some of this repeatability in local ancestry is linked to large minor parent ancestry "deserts" that coincide with known hybrid incompatibilities. Using wavelet analysis [38], we find the overall correlation in ancestry between *X. birchmanni* × *X. cortezi* hybrid populations is dominated by broad genomic scales, consistent with strong selection shortly after hybridization, and that there is likely a high density of selected sites. Moreover, repeatability in *X. birchmanni* × *X. cortezi* hybrid populations greatly exceeds what is observed in hybrid populations between the more closely related species *X. birchmanni* × *X. malinche*, pointing to pronounced changes in reproductive isolation with modest increases in genetic divergence (Fig 1B). This unique system with replicated hybridizing populations in 2 closely related species pairs gives us unprecedented power to unravel the dynamics of selection after hybridization and its impacts on repeatability in genome evolution.

## Results

### Chromosome scale genome assembly for *X. cortezi*

We generated a nearly chromosome-scale de novo assembly for *X. cortezi* using PacBio HiFi long-read sequencing at approximately 100× coverage. The genome was highly contiguous, with a contig N50 of 28,997,520 bp. Following reference-guided scaffolding to previously generated chromosome-level *X. birchmanni* and *X. malinche* assemblies (NCBI assembly id: GCA_036418095.1), the final *X. cortezi* assembly was chromosome-level with a scaffold N50 of 32,220,398 bp, and >99.4% of all sequence contained in the largest 24 scaffolds (corresponding to the 24 *Xiphophorus* chromosomes). The total assembled sequence length of 723 Mb is similar to other *Xiphophorus* assemblies and close to the expected length for this species based on previously collected flow cytometry estimates [59]. The *X. cortezi* genome was also highly complete, with 98.6% of actinopterygii BUSCOs present and in single copy (C:98.6%[S:97.0%, D:1.6%],F:0.4%,M:1.0%,n:3640), and the annotation process recovered a total of 25,032 protein coding genes (see Methods, Text A in S1 File). While the 2 genomes are largely syntenic, we also identified putative structural rearrangements between *X. birchmanni* and *X. cortezi* (Text A in S1 File and S1 and S2 Tables).

### Genome-wide ancestry in the Chapulhuacanito and Santa Cruz populations

Past work from our group has focused on hybridization between *X. birchmanni* and *X. cortezi* in the Santa Cruz river drainage [57]. While we collected samples from multiple sites in the Santa Cruz drainage in our previous work, our analyses suggested that hybrids at different sampling sites originated from the same hybridization event [57]. For simplicity, throughout the manuscript, we refer to samples collected at the Santa Cruz site as "Santa Cruz" and samples collected nearby (e.g., in historical collections) as samples from the "Río Santa Cruz." Here, we report a previously undescribed hybridization event between *X. birchmanni* and *X. cortezi* at the Chapulhuacanito population (21°12′10.58″N, 98°40′28.27″W) in the Río San

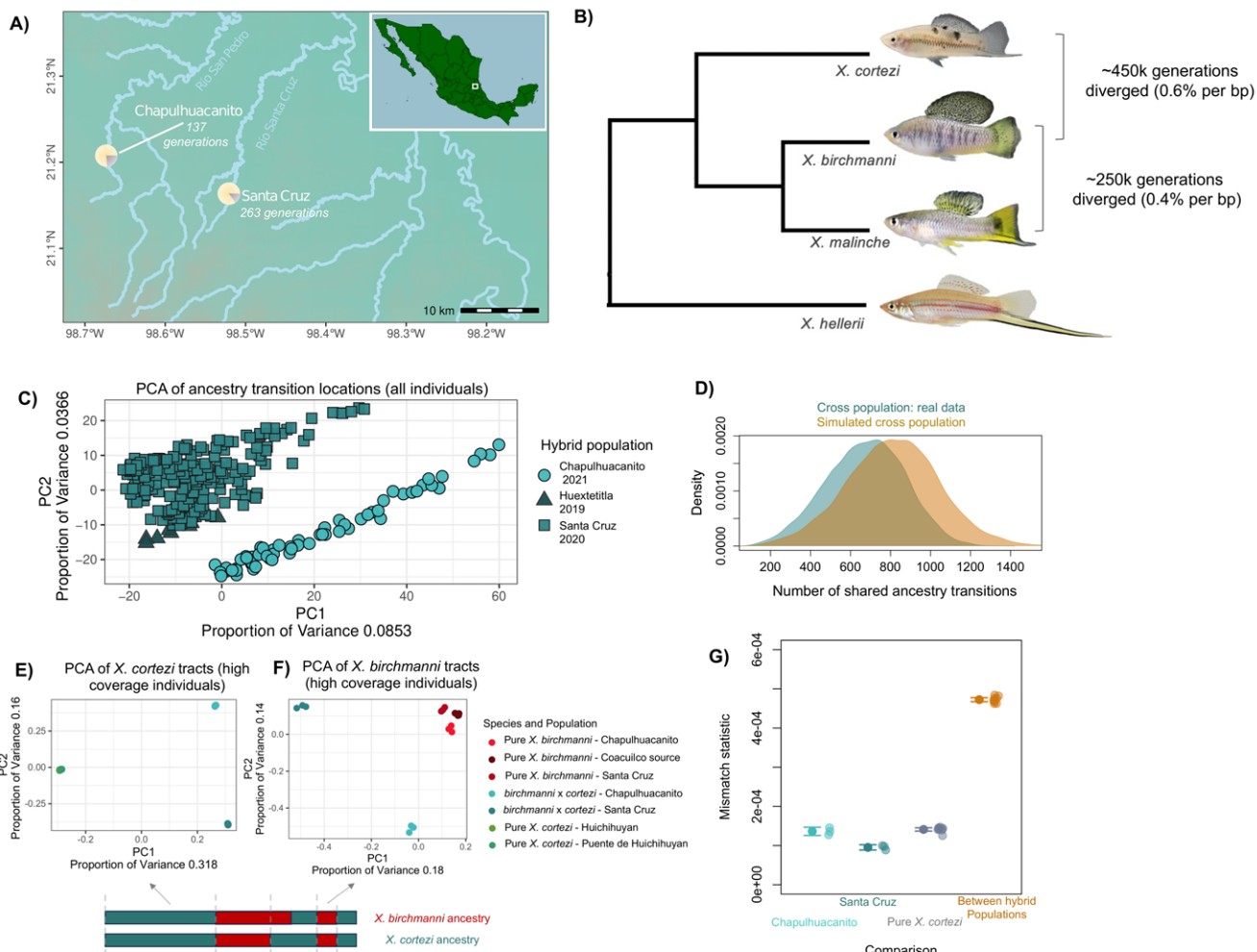

**Fig 1.** (**A**) Map of collection sites of *X. birchmanni* × *X. cortezi* hybrids in 2 different river drainages. Map was generated with borders and elevations from Natural Earth and using a custom shape file to delineate river locations. Pie charts show the average proportion of the genome hybrids derive from *X. cortezi* (yellow) and *X. birchmanni* (purple) in each population. Maximum a posteriori estimates of time since initial admixture are listed below each population name. Full results of ABC analyses of hybrid population demographic history can be found in S1 Fig and S3 Table. (**B**) Phylogenetic relationships between *X. birchmanni*, *X. malinche*, and *X. cortezi* and estimated divergence times between species from previous work. (**C**) PCA analysis of the locations of ancestry transitions indicates that the Santa Cruz and Chapulhuacanito populations have distinct recombination histories, while other individuals from the Santa Cruz drainage (the "Huextetitla" population) cluster with Santa Cruz. (**D**) Using simulations, we also find that the number of shared ancestry transitions across populations (i.e., cases where ancestry transitions occur in the same physical location along the genome) is comparable to that under a null model of independently evolving populations. Blue distribution shows the number of overlapping ancestry transitions across all pairs of individuals in Santa Cruz and Chapulhuacanito, and orange distribution shows the results of simulations of independent hybrid populations using the *X. birchmanni* recombination map (see Methods). Importantly, the shared ancestry transitions in the 2 populations do not exceed the number expected from simulations of independent population histories. (**E, F**) We also evaluated patterns of genetic variation using SNPs in high coverage individuals, subsetting the data to analyze tracts that are homozygous for *X. cortezi* (**E**) or *X. birchmanni* (**F**) in hybrid individuals. Schematic of diploid hybrid individual below the plots shows our approach for selecting regions for PCA analysis based on local ancestry in the 6 high-coverage hybrid individuals. Tracts from individuals in different hybrid populations separate from each other and the parental populations in PCA space (**E, F**). The sympatric *X. birchmanni* populations (**F**) found in both sites are genetically distinguishable from each other and the Coacuilco reference population but modestly so. See Text E in S1 File for a more in-depth discussion of these results. (**G**) Results of a "mismatch" analysis for comparisons of *X. cortezi* ancestry tracts within the 6 high coverage hybrid individuals and in pure *X. cortezi* source populations. We counted the number of sites within *X. cortezi* ancestry tracts where pairs of individuals from Santa Cruz and Chapulhuacanito were homozygous for different SNPs and divided this number by the total number of sites that passed our quality thresholds in each comparison (see Methods). We found striking differences in this mismatch statistic comparing individuals within population versus between populations. We repeated the same analysis within the 2 *X. cortezi* populations on the Río Huicihuyan for comparison. Differences between the Santa Cruz and Chapulhuacanito populations greatly exceed expectations for populations with shared demographic histories (Text C in S1 File). Semi-transparent points show the results of each comparison, bars and whiskers show the mean ± 2 standard errors. The data underlying this figure can be found in Dryad repository doi:10.5061/dryad.qnk98sfq1.

Pedro drainage, 17 km away by land and 130 km away in river distance from the Santa Cruz population (Fig 1A). While on average populations from the Santa Cruz drainage derive 85% to 89% of their genomes from the *X. cortezi* parental species, the Chapulhuacanito population is more admixed, with 76% of the genome derived from *X. cortezi* on average (Fig 1A). In both populations, *X. birchmanni* is the minor parent species.

We also sequenced historical samples from the Chapulhuacanito and Río Santa Cruz populations from 2003, 2006, and 2017. This sampling period spans approximately 40 generations based on reported generation times for this species group [60]. Since hybridization began in these populations more than a hundred generations before the present (see below), our earliest sampling points only survey the latest chapter in the history of *X. birchmanni* × *X. cortezi* hybrid populations. Theory predicts that in the first several generations following hybridization, admixture proportions can change dramatically due to selection [20,34,37], but after this initial period, change in genome-wide average ancestry is expected to slow dramatically [37,38]. The observed patterns in our data sets are concordant with these predictions. Genome-wide average ancestry was essentially unchanged from 2003 to recent sampling from 2019 to 2021 (Chapulhuacanito: 78 ± 1.2% *X. cortezi* in 2003 and 76 ± 2% *X. cortezi* in 2021; Río Santa Cruz: 87 ± 5% *X. cortezi* in 2003 and 88 ± 1% *X. cortezi* in 2019 to 2020).

## Demographic history of the hybrid populations

The demographic history of each hybrid population is also expected to impact how repeatable the outcomes of selection are and should be explicitly incorporated into analyses. When comparing repeatability in local ancestry across hybrid populations that formed on different timescales, both the amount of time for selection to shift ancestry at target loci and for genetic drift to shift ancestry at neutral loci would be expected to impact results. To incorporate these sources of variation into our analyses, we used an approximate Bayesian computation approach to explore the likely demographic histories of both the Santa Cruz and Chapulhuacanito populations (see Methods; [23]). We performed simulations drawing from uniform distributions of time since admixture, admixture proportion, and hybrid population size and log uniform distributions for migration rates from each parental species, using SLiM ([61], see Methods). We used the observed genome-wide admixture proportion, coefficient of variance in genome wide and local ancestry, and median ancestry tract length as summary statistics, and used ABCreg ([62]; see Methods) to infer posterior distributions for the time since admixture, admixture proportion, hybrid population size, and migration rates from each parental species, in both hybrid populations. While we did not recover well-resolved posterior distributions for hybrid population size for either population, we do recover well-resolved posterior distributions for other demographic parameters. Based on the maximum a posteriori (or MAP) estimate of these distributions, we find that hybridization began over a hundred generations ago in both drainages (Figs 1A and S1; Chapulhuacanito = 137; Santa Cruz = 263; see S3 Table for 95% confidence intervals), and that the migration rate from the parental populations has been very low (MAP estimate for Santa Cruz: $m_{cortezi} = 4 \times 10^{-5}$, $m_{birchmanni} = 0.00028$; Chapulhuacanito: $m_{cortezi} = 4 \times 10^{-5}$, $m_{birchmanni} = 0.00017$; S1 Fig; see S3 Table for 95% confidence intervals). In all subsequent simulations described in the manuscript, we explicitly incorporate this inferred demographic history to build our expectations of cross-population correlations under neutrality or under different models of selection.

## Confirming the independent origin of the 2 hybrid populations

Given substantial river distance between the 2 *X. birchmanni* × *X. cortezi* hybrid populations (approximately 130 km; Fig 1A) and the limited dispersal capabilities of these small fish

([63,64]; Text B in S1 File), we had good reason to believe that the 2 populations originated independently. However, because of the extraordinarily high correlations in local ancestry we observed across the 2 populations (see below), we sought additional evidence that they were completely independent in origin.

Since we inferred local ancestry for individuals from both populations, we have access to information about historical recombination events in these populations. Specifically, a subset of recombination events that occurred in the hybrid ancestors of present-day individuals will be detectable as ancestry transitions along the chromosomes of present-day individuals. We tested for potential overlap in the locations of ancestry transitions across populations, which could indicate a period of shared history following hybridization. We generated a matrix containing the locations of ancestry transitions in each hybrid individual in our dataset (see Methods) and performed a principal component analysis. We see that the Santa Cruz and Chapulhuacanito populations separate out in PC space in this analysis (Fig 1C). This suggests that the 2 populations have largely distinct recombination histories in the recent past. We also find that the frequency at which the locations of ancestry transitions are shared between individuals in the Santa Cruz and Chapulhuacanito populations is similar to the frequency expected under a null model (Fig 1D), again pointing to independent population histories.

We further explored these patterns using high-coverage whole genome sequencing data of 3 individuals from sympatric *X. birchmanni* populations at both Santa Cruz and Chapulhuacanito, and 3 naturally occurring hybrids at the 2 sites. We called variants (see Methods) and performed principal component analysis on sympatric *X. birchmanni* individuals, hybrid samples, and pure *X. birchmanni* and *X. cortezi* collected from allopatric populations (S2–S4 Figs). Moreover, we performed local ancestry inference on the natural hybrids for which we had generated deep-sequencing data and identified homozygous *X. birchmanni* and homozygous *X. cortezi* ancestry tracts within these individuals (limiting our analysis to tracts that were the same ancestry state in all 6 deep-sequenced hybrids). We extracted these regions from the natural hybrids and from the parental genomes and performed principal component analysis on regions of *X. birchmanni* and *X. cortezi* ancestry separately. We found that ancestry tracts derived from the 2 hybrid populations formed separate clusters (Fig 1E and 1F) and individuals from the 2 hybrid populations have high levels of sequence mismatch, compared to the levels observed within populations (Fig 1G; Methods). Simulations indicate that the degree of sequence mismatch observed across the 2 hybrid populations (Fig 1G) greatly exceeds what would be expected if these populations separated recently (i.e., in the ~200 generations since initial hybridization; S5 Fig; Text C in S1 File).

We next used direct measures of relatedness to search for evidence of close relatives across different populations, which would be indicative of unexpected connectedness between them. For each ancestry source we estimated a genetic relatedness matrix [65] using only regions homozygous for that ancestry source from high-coverage hybrid individuals. We find no evidence of close relatedness across hybrid populations in these regions (see Methods; S6 Fig). We also used IBDseq [66] to evaluate evidence of shared IBD tracts within and between populations in our high-coverage data (although we note that we have limited power given our sample size; see Methods and Text D in S1 File). While we found strong evidence of IBD tracts [24] within both hybrid populations and within *X. birchmanni* and *X. cortezi* populations (S7 and S8 Figs and S4 Table), we found very limited evidence for IBD tracts shared across the 2 hybrid populations (S7 Fig). Together, these results corroborate our expectations from geographic isolation and demographic analyses, indicating that hybrid populations in the Santa Cruz and Chapulhuacanito rivers have originated and evolved independently. See Text D and E in S1 File for a more thorough discussion of the implications of analyses of relatedness and genetic variation within and between populations.

## Correlations between minor parent ancestry, the local recombination rate, and the density of coding and conserved basepairs

Past work on hybrid populations of *Xiphophorus* and in other systems [3,19,30,32,35,57] has found that the frequency of minor parent ancestry in the genome often correlates with factors such as the local recombination rate and the density of functional basepairs (e.g., coding regions). In the presence of selection against minor parent ancestry (due to hybrid incompatibilities or other mechanisms; [20,34]), both theory and simulations [37] predict that the level of minor parent ancestry will be positively correlated with the local recombination rate. Similarly, if selected sites fall more frequently in coding (or conserved) regions of the genome, and selection is sufficiently polygenic, we might expect to see a depletion of minor parent ancestry in these regions.

We tested for correlations between the local recombination rate estimated in *X. birchmanni* and local ancestry along the genome in a range of window sizes in both Chapulhuacanito and Santa Cruz (Fig 2A and S5 Table). Although we have developed recombination maps for both species (see Methods), we chose to use the *X. birchmanni* map because it is likely to be more accurate (see [35]; Text F in S1 File) and our analyses suggest that it is extremely similar to the *X. cortezi* map (S9 and S10 Figs and Text F in S1 File). Regardless of window size, we observe strong positive correlations between the local recombination rate and average minor parent ancestry in both populations (Fig 2A and S5 Table). After controlling for the strong effects of local recombination rate, we find that the density of coding (and conserved) basepairs also correlates with the distribution of minor parent ancestry in Chapulhuacanito and Santa Cruz (S6 and S7 Tables; see also [57]). In particular, regions of the genome with an especially high density of coding (or conserved) basepairs appear to be depleted in minor parent ancestry (Fig 2B).

## Repeatability in local ancestry between replicate hybrid populations

We found that local ancestry along the genome was surprisingly repeatable across the 2 *X. birchmanni* × *X. cortezi* hybrid populations (Fig 2C). That is, the observed minor parent ancestry in a given 100 kb region of the genome in one population was highly predictive of the observed minor parent ancestry in that same region in the other population (Spearman's $\rho$ = 0.79; $p = 2 \times 10^{-171}$). We note that because adjacent windows are not independent, for all analyses we report $p$-values after thinning data to include only 1 window per Mb (admixture LD in both populations decays to background levels over this distance; S11 Fig). The observed correlations in ancestry across populations exceed what we have previously detected in replicate *X. birchmanni* × *X. malinche* hybrid populations (Fig 2D). While we detected these patterns across window sizes, they generally increased with larger window sizes (S8 Table). We find that these correlations are robust to controlling for shared features of genome architecture like the local recombination rate and the locations of coding and conserved basepairs using a partial correlation approach (S9 Table; see Methods).

Simulations incorporating the inferred demographic history of the 2 populations indicate that this strong cross-predictability is not expected under neutrality but can be produced in scenarios of hybridization followed by strong selection on many loci (Text G in S1 File). This suggests that repeatability in minor parent ancestry across *X. birchmanni* × *X. cortezi* hybrid populations is driven by a shared architecture of selection on hybrids. For comparison, we evaluated correlations in local ancestry observed when subsampling individuals from the same population and sampling year, samples from the same populations but different sampling years, and populations sampled from different sites on the same river. We reasoned that for each of these comparisons, samples are expected to largely share the same demographic history

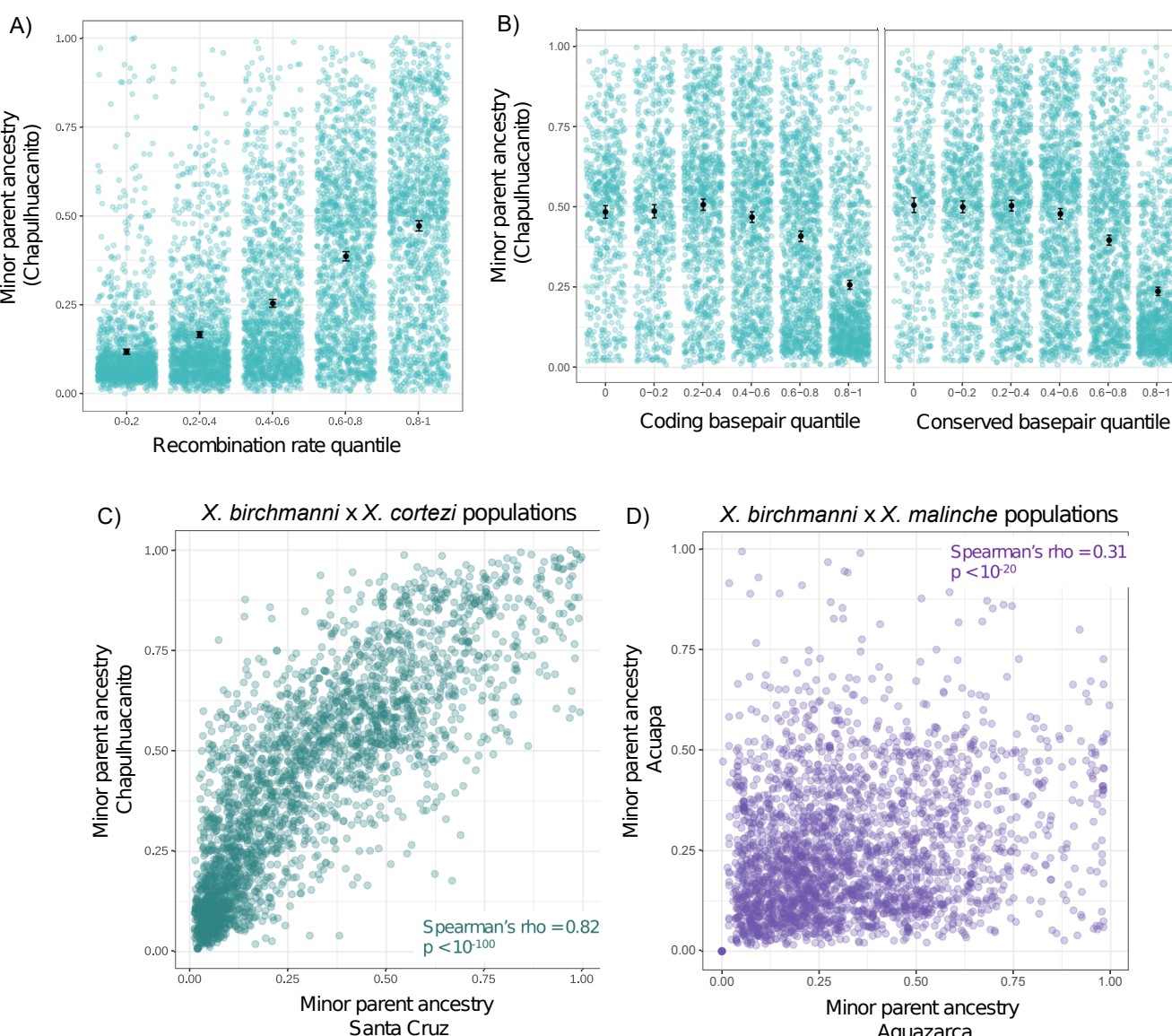

**Fig 2.** (**A**) Minor parent ancestry in the Chapulhuacanito population is strongly correlated with the local recombination rate. Here, ancestry and recombination are summarized in 250 kb windows (see also Fig 3C for wavelet-based analysis). (**B**) After accounting for the strong effect of recombination rate by summarizing ancestry in 0.25 cM windows, we also find that minor parent ancestry is depleted in regions of the genome linked to large numbers of coding or conserved basepairs. We previously reported similar results for the Santa Cruz population for both recombination rate and functional basepair density [57] and for *X. birchmanni* × *X. malinche* hybrid populations [35]. (**C**) Average minor parent ancestry is strikingly correlated across the Santa Cruz and Chapulhuacanito populations. Shown here are analyses of 0.5 cM windows (Spearman's $\rho = 0.82$, $p < 10^{-100}$); these results are observed across all spatial scales tested in both physical and genetic distance (see S12 Fig and S8 and S9 Tables). (**D**) By contrast, minor parent ancestry is substantially less correlated between 2 *X. birchmanni* × *X. malinche* hybrid populations. Shown here are analyses of 0.5 cM windows (Spearman's $\rho = 0.31$, $p < 10^{-20}$). For additional comparisons of ancestry in *X. birchmanni* × *X. malinche* hybrid populations, see [35,57]. The data underlying this figure can be found in Dryad repository doi:10.5061/dryad. qnk98sfq1.

and history of selection. Reassuringly, we found that correlations in these analyses greatly exceeded those observed between Chapulhuacanito and Santa Cruz (S12 Fig and S8 Table).

Our simulations indicate that the striking correlations we see in local ancestry across Chapulhuacanito and Santa Cruz (Fig 2C) could be driven by a shared architecture of selection on hybrids in these populations (see below, Text G in S1 File). However, we wanted to thoroughly

rule out other possible explanations, namely that technical factors might contribute to this signal. These approaches are described in detail in the Methods and S1 File, but we discuss them briefly here. We used simulations and analyses of lab generated crosses to confirm that our local ancestry inference approach is highly accurate (S13–S15 Figs; Methods and Text H in S1 File). We used simulations to artificially induce high error rates in local ancestry inference and found that it could not generate the patterns observed in our data (Text H in S1 File). We repeated analyses removing regions that are prone to error in local ancestry inference (S10 Table; Text I in S1 File) and controlling for our power to infer local ancestry along the genome (see Methods; S10 Table), among other analyses (see S10 Table; Methods; Text I in S1 File). None of these analyses qualitatively changed our results (Text G–I in S1 File).

Simulations indicate that it is possible for selection alone to drive cross-population correlations at the magnitude we infer in Santa Cruz and Chapulhuacanito in scenarios where selection acts on many loci and is exceptionally strong (average *s* drawn from an exponential distribution of 0.4–0.6; Text G in S1 File). Our results from lab-generated *X. birchmanni* × *X. cortezi* hybrids indicate that hybrids in this cross suffer immense fitness consequences, suggesting that such strong selection is plausible (see Discussion; [67]). Indeed, evaluating patterns of local ancestry across the 2 independently formed populations, we can see evidence for large, shared deserts of minor parent ancestry (Fig 3A). This hints that the correlations we observe in our data may be largely driven by strong selection acting shortly after hybridization, resulting in shared patterning of minor ancestry over broad spatial scales along the genome. To evaluate this question in more depth and across spatial scales in the genome, we next used a wavelet-based analysis of cross population ancestry correlations [38].

## Wavelet transform approach to infer the spatial scale of correlations in ancestry

In our windowed analyses, the correlations in ancestry between the Santa Cruz and Chapulhuacanito populations increase as we consider larger window sizes, suggesting that the observed correlations are driven by covariation in ancestry at large genomic scales (S8 Table). Similarly, we find that the correlations between recombination rate and minor parent ancestry become stronger in larger genomic windows (S5 Table).

Theory predicts that the strength of selection on hybrids will vary dramatically over time, since the removal of ancestry tracts harboring alleles that are deleterious in hybrids will be most rapid in the earliest generations following hybridization when ancestry tracts are long [34,36,37]. Furthermore, these dynamics can establish spatial ancestry patterns along the genome that persist over time and constrain subsequent evolution. This leads to the prediction that the genomic scale of autocorrelation in ancestry will be informative about the timing and strength of selection (relative to the onset of hybridization) [38]. To better understand the role of selection in shaping genomic ancestry patterns across replicate hybrid populations, we applied recently developed methods based on the Discrete Wavelet Transform [38] to our data (see Methods; Text J in S1 File). The intuition behind this analysis is as follows: moving along a chromosome, the ancestry proportion deviates around its chromosome-wide average, and this variation occurs over a range of different spatial scales (genomic window sizes, roughly speaking). The wavelet transform can be used to summarize the scales of variance in ancestry along a chromosome, as well as the contributions of each scale to the overall correlation between 2 signals measured along the chromosome (e.g., ancestry and recombination), where each component carries independent information about the overall correlation (see Methods). Because the scale of variation is ultimately determined by the lengths of admixture tracts, these signals contain information about the timing of selection and drift relative to the onset of hybridization [38].

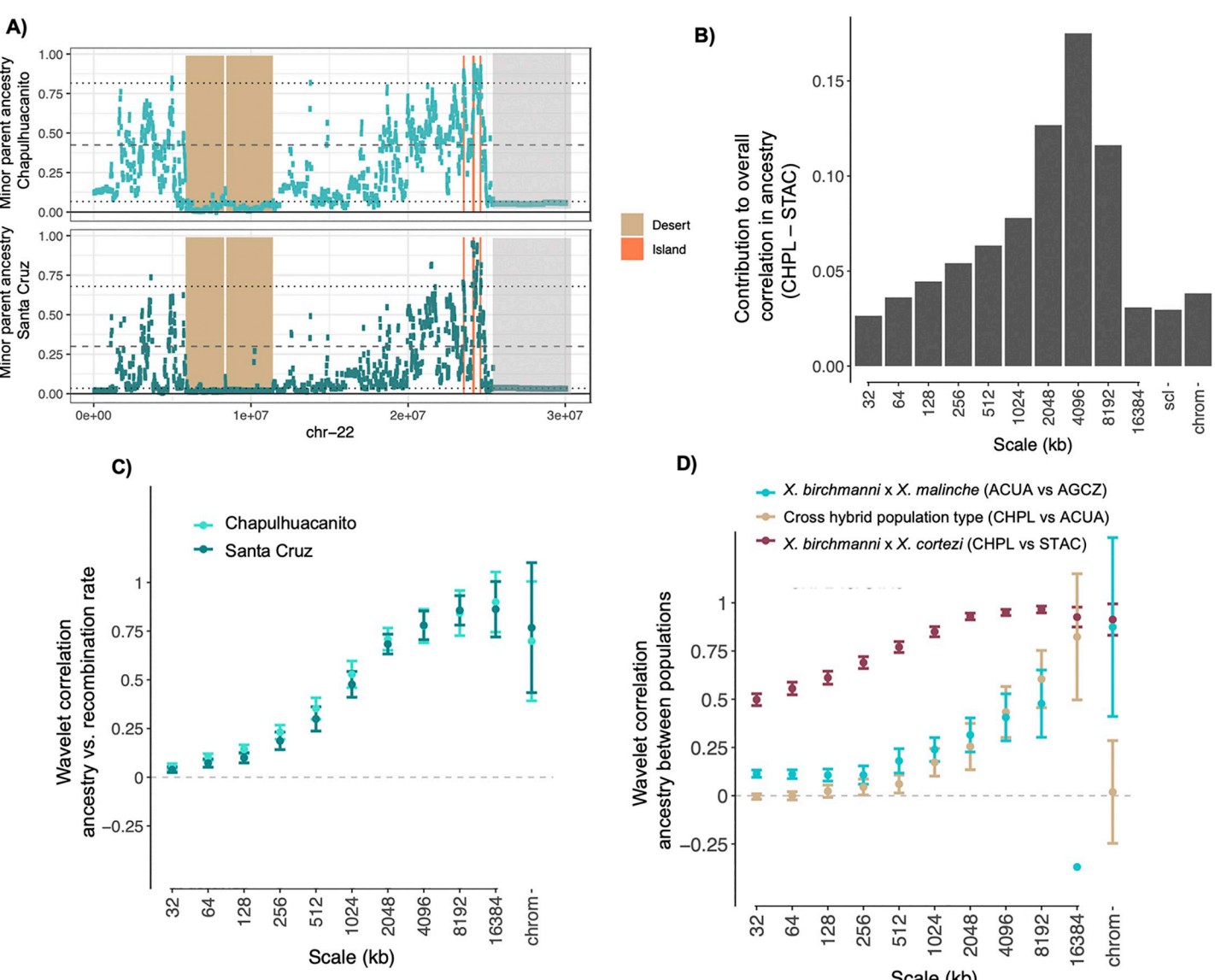

**Fig 3.** (**A**) Example of large shared minor parent ancestry deserts identified on chromosome 22 (tan) as well as shared minor parent ancestry islands (peach) in Chapulhuacanito and Santa Cruz. Note also large regions of low minor parent ancestry at approximately 25–30 Mb found across both populations that do not pass the threshold for being designated as shared ancestry deserts (light gray; in this case the region exceeds the 5% threshold for Santa Cruz). Dashed lines indicate the average ancestry genome-wide and dotted lines represent lower and upper 10% quantiles of minor parent ancestry. (**B**) Spatial wavelet decomposition of the overall Pearson correlation between inferred minor parent ancestry in Chapulhuacanito vs. Santa Cruz (CHPL vs. STAC) measured at a resolution of 32 kb. The contribution of a given spatial scale is a weighted correlation of wavelet coefficients for the 2 signals at that scale, weighted by the portion of the total variance attributable to that scale (see Methods). Correlations among chromosome means also contribute (chrom), as well as a leftover component (scl) due to irregularity of chromosome lengths. (**C**) Wavelet correlations between inferred minor parent ancestry and recombination rate for both Chapulhuacanito and Santa Cruz populations. Note that here correlations at each scale are not weighted by variances at the corresponding scales. Points are weighted averages across chromosomes with error bars representing 95% jackknife confidence intervals. (**D**) Wavelet correlations between inferred minor parent ancestry proportion in cross population comparisons between hybrids derived from the same hybridizing pair (CHPL vs. STAC—*X. birchmanni* × *X. cortezi*; ACUA vs. AGCZ–*X. birchmanni* × *X. malinche*) and from different hybridizing pairs (CHPL—*X. birchmanni* × *X. cortezi* vs. ACUA—*X. birchmanni* × *X. malinche* hybrids at the Acuapa site). For visualization, we omit the confidence interval for the wavelet correlation of ancestry in the 2 *X. birchmanni* × *X. malinche* populations (ACUA vs. AGZC) at the largest scale, since it spans nearly from −1 to 1. Note that the identity of the minor parent species differs across hybrid population types (*X. birchmanni* in Chapulhuacanito and Santa Cruz and *X. malinche* in Acuapa and Aguazarca). The data underlying this figure can be found in Dryad repository doi:10.5061/dryad.qnk98sfq1.

Using this approach, we found that the overall correlation between minor parent ancestry and recombination in both replicate populations is predominantly attributable to broad genomic scales (Fig 3C). Furthermore, wavelet correlations between minor parent ancestry and

recombination were strongly positive in replicate populations, with the strongest correlations observed at the broadest genomic scales (S16 Fig). The squared correlation coefficients for ancestry versus recombination can be interpreted as the percent of variance in ancestry at each scale attributable to genome-wide selection against minor parent ancestry [38], since these correlations are only generated by selection and not by drift (barring errors in ancestry inference; see Text J in S1 File). Applying this logic, correlations with recombination indicate that roughly 80% of the variance in ancestry at the broadest genomic scales (e.g., 16 Mb) in the Santa Cruz and Chapulhuacanito populations can be attributed to selection against minor parent ancestry. By contrast, comparatively little of the variance in ancestry at fine genomic scales is attributable to selection against minor parent ancestry (e.g., 0.2% at a scale of 32 kb).

We next applied this approach to investigate the correlation between minor parent ancestry across the 2 replicate *X. birchmanni* and *X. cortezi* populations. We found that across scales, cross-population ancestry correlations between *X. birchmanni* × *X. cortezi* hybrid populations were stronger than the correlations observed with recombination rate, especially at finer spatial scales. Thus, ancestry in a replicate hybrid population is a better predictor of fine-scale genetic ancestry patterns than recombination rate. This implies that recombination alone only captures a portion of the total effects of selection on ancestry patterns, and that its effects in mediating parallel genomic outcomes of hybridization manifest predominantly over broad genomic scales. From cross-population ancestry wavelet correlations, estimates of the proportion of ancestry variance attributable to selection on minor parent ancestry range from approximately 25% at a scale of 32 kb to as high as 93% at a scale of 8 Mb (Fig 3D). Surprisingly, we found that significant positive correlations persisted even at very small spatial scales (S17 Fig). This pattern is consistent with convergent selection shaping very fine scale ancestry patterns, although we discuss important caveats to this interpretation in Text J in S1 File. Nonetheless, the magnitude and scales of ancestry correlations across populations suggest that predictability is driven by both early and continued selection on hybrids.

For comparison, we repeated these analyses in 2 *X. birchmanni* × *X. malinche* hybrid populations, the Acuapa population and the Aguazarca population [35]. *X. birchmanni* and *X. malinche* are more closely related than *X. birchmanni* and *X. cortezi* (Fig 1B), and hybridization began more recently (within the last 50 to 100 generations; [23]). We again find strong positive correlations between minor parent ancestry in the 2 populations at broad genomic scales, but these are noticeably reduced relative to the cross-population comparison between the 2 *X. birchmanni* × *X. cortezi* populations (Figs 3D and S17 and Text J in S1 File). These results are consistent with weaker selection overall against minor parent ancestry in *X. birchmanni* × *X. malinche* hybrid populations, and/or fewer loci under selection, both of which may be expected given that these species diverged more recently (Fig 1B; [35,68]). Moreover, previous work analyzing wavelet correlations between minor parent ancestry and recombination rate in *X. birchmanni* × *X. malinche* populations found that only ~20% of the variation in minor parent ancestry at large spatial scales was attributable to selection [38]. Overall, these results suggest that genome evolution after hybridization is substantially more predictable for *X. birchmanni* × *X. cortezi* hybrids.

Finally, we examined the genomic scale of shared ancestry patterns between a *X. birchmanni* × *X. cortezi* hybrid population and a *X. birchmanni* × *X. malinche* hybrid population (the Chapulhuacanito and Acuapa populations, respectively). We observed positive correlations in minor parent ancestry at broad scales but find that these correlations are dramatically reduced at fine scales, especially compared to analyses of the populations of the same hybridizing pair (Fig 3D). This would be expected if replicate populations of the same hybridizing pair show greater overlap in the fine-scale targets of selection than populations of different hybridizing pairs. The positive fine-scale ancestry correlations within replicate hybrid populations

(Figs 3D and S17) are consistent with this interpretation (see also Text J in S1 File). We thus suggest that broad-scale predictability among different hybridizing pairs may be driven primarily by effects of shared genome architecture rather than shared identity of selected loci.

## Repeatability in minor parent deserts and islands between replicate *X. birchmanni* × *X. cortezi* populations

Given that the results of wavelet-based analyses point to shared targets of selection across *X. birchmanni* × *X. cortezi* hybrid populations, we were interested in whether we could identify individual loci that are likely to be under selection. Loci that are shared targets of selection could be alleles that are globally deleterious (or beneficial), or those that are involved in hybrid incompatibilities between *X. birchmanni* × *X. cortezi*. Using our large recent population samples, we identified contiguous regions of low minor parent ancestry, or minor parent ancestry "deserts," in each *X. birchmanni* × *X. cortezi* hybrid population and asked how frequently they overlapped across populations (see Methods). Simulations suggest that our approach has high sensitivity and low false positive rates (approximately 70% power at s = 0.05; average of 2 to 4 shared deserts detected genome wide in neutral regions; see Text K in S1 File). We identified 115 "deserts" of low minor parent ancestry in Santa Cruz and 152 deserts in Chapulhuacanito. Strikingly, 38 of these regions overlapped across the 2 populations, greatly exceeding expectations by chance (Fig 4A; $p < 0.001$ by simulation; see Methods). The average length of these regions was 1.8 Mb with a total of approximately 40 Mb of the 723 Mb genome falling into shared deserts. Since the typical ancestry tract length for *X. cortezi* (i.e., the major parent) in these populations is much smaller (approximately 150 kb), this hints that these regions may have changed in ancestry shortly after initial hybridization. These shared minor parent ancestry deserts are excellent candidates for shared regions under selection in the 2 hybrid populations.

Similarly, we identified regions of especially high minor parent ancestry in each *X. birchmanni* × *X. cortezi* hybrid population and asked how frequently they overlapped across populations compared to expectations by chance (see Methods). In doing so, we found evidence for 89 shared minor parent "islands" out of 238 islands in Santa Cruz and 147 in Chapulhuacanito, again exceeding the level of sharing expected by chance (Fig 4A; $p < 0.001$ by simulation; Methods). The typical length of shared islands was 190 kb, much smaller than that observed for shared deserts, but together these regions still covered a substantial portion of the genome (approximately 29 Mb). We report the genes observed in these regions (S11 Table) and analysis of functional enrichment in the supplementary materials (Text L in S1 File).

We compared minor parent deserts and islands identified in the *X. birchmanni* × *X. cortezi* hybrid populations to those detected in the *X. birchmanni* × *X. malinche* hybrid populations. As expected, we found many fewer shared deserts and islands across hybrid population types (S18 Fig), but shared deserts and islands exceeded expectations by chance in most comparisons (most comparisons $p < 0.001$ by simulation, with a maximum of $p = 0.11$).

Since we had access to time-series data for both the Santa Cruz and Chapulhuacanito populations, we were interested in evaluating how ancestry at minor parent deserts and islands has changed over the last 40 generations. Given that both hybrid populations are estimated to be over 100 generations old, we would expect that loci under strong or moderate selection would be fixed even at the earliest time points in our data set. Indeed, we find that regions that fall into shared ancestry deserts tend to have low minor parent ancestry in 2003 and maintain low ancestry through time (Fig 4B and 4D). The same is generally true for regions of high minor parent ancestry, although we do identify 6 minor parent islands where minor parent ancestry significantly increases between 2003 and 2020–2021 (S11 Table).

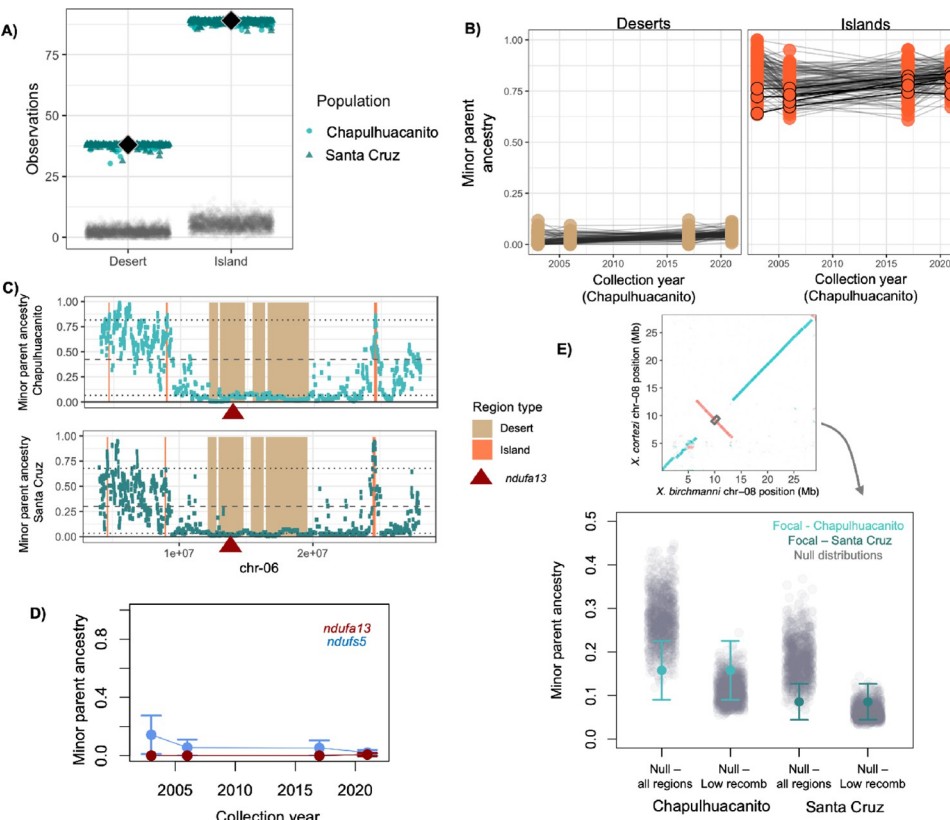

**Fig 4.** (**A**) Shared minor parent ancestry deserts and islands in *X. birchmanni* × *X. cortezi* populations (Chapulhuacanito and Santa Cruz—colored points) occur at a much higher rate than expected by chance (gray points, see methods). Black diamonds show the observed number of shared minor parent deserts or islands across the 2 populations and colored points show the results of jack-knife bootstrapping the data from each population in 10 cM blocks (circles–bootstrap results from Chapulhuacanito, triangles–bootstrap results from Santa Cruz). (**B**) Ancestry of shared minor parent deserts (left) and islands (right) through time in the Chapulhuacanito data set. Points show ancestry at individual deserts or islands for each sampling year, and lines connect results for a given desert or island across years. Shared minor parent ancestry deserts were largely fixed by the onset of genetic monitoring of these populations approximately 40 generations ago. Islands also tended to have high minor parent frequency at the onset of sampling, but several islands do change significantly in minor parent ancestry over the sampling period (S11 Table). Islands that increase significantly in minor parent ancestry through time are outlined in black. (**C**) One shared minor parent ancestry desert on chromosome 6 overlaps with a known mitonuclear incompatibility generated by combining the *X. cortezi* mitochondria with homozygous *X. birchmanni* ancestry at *ndufa13* [58,69]. Local ancestry along chromosome 6 in Chapulhuacanito is shown in the top plot and local ancestry along chromosome 6 in Santa Cruz is shown in the bottom plot. The locations of shared deserts and islands are highlighted in tan and peach, respectively. The location of *ndufa13* is indicated by the red triangle. Dashed lines indicate the average ancestry genome-wide and dotted lines represent lower and upper 10% quantiles of minor parent ancestry. (**D**) Both *ndufa13* and another gene involved in mitonuclear incompatibility between *X. cortezi* and *X. birchmanni*, *ndufs5*, are nearly fixed for major parent ancestry at the onset of our time-series sampling. (**E**) With our new long-read reference assemblies, we evaluated minor parent ancestry at the center of inversions (focusing on inversions >100 kb) that differentiated *X. birchmanni* and *X. cortezi* in the 2 hybrid populations. Example alignment of a large inversion identified on chromosome 8 is shown in the inset. For each inversion, we sampled ancestry in a 50 kb window that overlapped with the center of the inversion (schematically shown by the gray rectangle in the inset). We found that minor parent ancestry was modestly depleted in the 2 hybrid populations (colored points and whiskers) at inversions compared to randomly sampled regions of the genome (gray points–"all regions"). However, when we generated null data sets only from regions of the genome with low recombination rates (lowest 5% quantile of recombination rate), we found that inversions did not show unusually high depletion of minor parent ancestry. This suggests that depletion of minor parent ancestry at inversions may be driven by reduced recombination in these regions in hybrids. The data underlying this figure can be found in Dryad repository doi:10.5061/dryad.qnk98sfq1.

Finally, we evaluated ancestry at rearrangements identified between *X. birchmanni* and *X. cortezi* based on our new PacBio HiFi-based assemblies. We identified 9 inversions greater than 100 kb, ranging in size from 218 kb to 6.7 Mb (S2 Table and S19 Fig). These inversions were concentrated on chromosomes 8 and 17 (6 out of 9 of the inversions). As chromosomal inversions tend to suppress recombination in heterozygotes, we predicted that these regions would be especially depleted in minor parent ancestry. Notably, we found that on average these regions were depleted in minor parent ancestry compared to expectations from null simulations (Fig 4E), but not when compared to non-inverted regions of the genome that had exceptionally low recombination rates (Fig 4E).

## Ancestry at known incompatibilities identified between *X. birchmanni* and *X. cortezi*

We were also interested in evaluating patterns of minor parent ancestry locally at regions that are known to be under selection in hybrids between *X. birchmanni* and *X. cortezi*. Other work from our lab has identified a mitonuclear hybrid incompatibility between individuals with the *X. cortezi* mitochondria and homozygous *X. birchmanni* ancestry at *ndufs5* and *ndufa13* [57,58,67]. $F_2$ hybrids that inherit the *X. cortezi* mitochondrial haplotype and 2 copies of the *X. birchmanni* allele at *ndufs5* experience mortality during embryonic development [58,69]. Inheriting the *X. cortezi* mitochondrial haplotype and 2 copies of the *X. birchmanni* allele at *ndufa13* causes higher rates of postnatal mortality. Because hybrid populations at both Santa Cruz and Chapulhuacanito have fixed the *X. cortezi* mitochondrial haplotype (S12 Table; [57,58]), this leads to the strong expectation that they will largely have purged *X. birchmanni* ancestry at *ndufs5* and *ndufa13*.

We evaluated ancestry in these regions of the genome in our large sample of hybrid individuals from both Santa Cruz and Chapulhuacanito. We identified a large, shared ancestry desert surrounding *ndufa13* on chromosome 6 (Fig 4C). For *ndufs5*, the region surrounding the gene on chromosome 13 was identified as an ancestry desert in Chapulhuacanito, but not in Santa Cruz. Closer examination of this region (S20 Fig) indicates that *X. birchmanni* ancestry at *ndufs5* is depleted in Santa Cruz but falls just above the 5% quantile of minor parent ancestry used to identify deserts genome-wide in Santa Cruz (the 5% quantile was 2.2% *X. birchmanni* ancestry while an average of 2.3% *X. birchmanni* ancestry was observed at *ndufs5;* see Methods). Moreover, both regions were consistently low in *X. birchmanni* ancestry through time in our samples from Chapulhuacanito (Fig 4D) and no individuals homozygous for *X. birchmanni* ancestry at either region were observed across the 2 populations. Based on predictions from Hardy–Weinberg equilibrium <0.05% of mating events would be expected to produce embryos incompatible at *ndufs5* or *ndufa13* in either population. Since these 2 genes form part of mitochondrial protein complex I, we also analyzed ancestry at genes that are involved in protein complexes genome wide (S21 Fig and Text M in S1 File).

Results from a recent study identified 2 additional regions on chromosomes 7 and 14 that impact viability in $F_2$ hybrids between *X. birchmanni* and *X. cortezi* [67]. While these regions are currently poorly resolved given the size of the mapping panel and the use of early generation hybrids (where ancestry tracts are large; [67]), we were curious if they overlapped with any minor parent ancestry deserts in Santa Cruz or Chapulhuacanito. We found that the region on chromosome 14 overlaps with a large shared ancestry desert and the region on chromosome 7 overlaps with an ancestry desert found in Santa Cruz (S22 Fig). We note, however, that these findings are not necessarily surprising given the size of the mapped regions that impact viability in $F_2$ hybrids [67]. For example, the region identified on chromosome 14 in $F_2$ hybrids is approximately 6 Mb and has a 27% probability of overlapping with a shared desert by chance based on simulations.

## Discussion

The extent to which genome evolution after hybridization is predictable is an open question in evolutionary biology. Given the large number of species that have exchanged genes with their close relatives, the answer to this question has wide ranging implications for understanding the genomes of species across the tree of life. While the question of whether evolution is repeatable is one of the core questions in evolutionary biology, few studies have been able to address this in the context of hybridization. This is likely because addressing it requires access to multiple, independently formed hybrid populations and accurate local ancestry inference approaches where technical factors such as variation in error rates or power to infer ancestry along the genome can be excluded as drivers of the observed patterns. Even well-studied cases with excellent genomic resources such as the human-Neanderthal and human-Denisovan admixture events present a challenge in appropriately accounting for such technical factors.

Here, we further developed *Xiphophorus* as a natural biological system in which to address these fundamental questions. We describe 2 hybrid populations between *X. cortezi* and *X. birchmanni* that formed in different river drainages in the last ~150 to 300 generations. Multiple lines of evidence—from geography to genetic variation to recombination history—confirm that the 2 hybrid populations formed independently. *X. cortezi* and *X. birchmanni* diverged an approximately 450k generations ago [68] and we estimate pairwise sequence divergence at 0.6%. Since levels of within-species polymorphism are relatively low, this results in a high density of fixed ancestry informative sites—approximately 4 per kb—with which to precisely infer ancestry along the genome and compare ancestry variation across the 2 populations.

Shortly after hybridization, hybrid genomes may contain large numbers of selected alleles that are linked on the same haplotype. Accordingly, both theory and empirical results have indicated that selection interacts with the global and local recombination rate to reshape minor parent ancestry in the genome (assuming that minor parent ancestry is on average deleterious; [35–37]). As in previous studies of *Xiphophorus* hybrids [17,35,57,58], we find a strong depletion of ancestry from the minor parent species (*X. birchmanni* in both populations) in regions of the genome with low recombination rates (Fig 2A), as well as a more subtle depletion of minor parent ancestry in regions of the genome of high coding (or conserved) basepair density (Fig 2B). Moreover, wavelet analyses indicate that correlations between minor parent ancestry and recombination rate are primarily driven by the broadest spatial scales (i.e., >4 Mb; Figs 3C and S12), suggesting that selection on early generation hybrids is driving patterning of minor parent ancestry at a genome-wide scale in both populations [38]. These analyses suggest that a striking amount of local ancestry variation at broad spatial scales is attributable to the action of natural selection (approximately 80%).

Perhaps the most surprising result of our study is the extraordinarily high correlations in local ancestry across the 2 *X. cortezi* and *X. birchmanni* hybrid populations (Fig 2C). The results of wavelet analyses indicate that broad-scale changes in ancestry along the genome in one hybrid population (at the scale of >8 Mb) predict a remarkable ~90% of the variance in the other hybrid population. We found that this cross-population repeatability was robust to iterations of the analysis controlling for potential technical confounders (see Methods; S10 Table). Since shared patterns of ancestry deviations are not predicted under neutrality, these results demonstrate that the correlations we observe are attributable to natural selection driving parallel changes in minor parent ancestry in the 2 hybrid populations, presumably due to selection on the same loci. Since these correlations are strongest at the broadest spatial scales in the genome, this indicates that natural selection acting shortly after hybridization was important in establishing them. The degree of cross-population repeatability we observe here exceeds that reported in other studies that have found evidence for such patterns [29,38,56,57].

What mechanisms could drive such high repeatability in minor parent ancestry across independently formed hybrid populations? Given the frequency of hybrid incompatibilities in *Xiphophorus* [17,35,58] and the fact that neither *X. birchmanni* or *X. cortezi* have experienced sustained bottlenecks like those observed in other *Xiphophorus* species ([35,68]; S4 Fig), we predicted that selection on hybrid incompatibilities may be an important driver of this signal. In simulations, we confirmed that strong selection on the same hybrid incompatibilities can, in principle, generate exceptionally high correlations in local ancestry across populations, similar to those observed in our data (Text G in S1 File; S23 and S24 Figs). Results from artificial crosses between *X. cortezi* and *X. birchmanni* support the conclusion that selection is extremely strong on early-generation hybrids. One $F_1$ cross direction fails to develop (with *X. birchmanni* mothers) and the other produces offspring with a 6:1 male sex-bias (with *X. cortezi* mothers; [67]).

In the case of strong selection against intrinsic hybrid incompatibilities, we expect to see large "deserts" of minor parent ancestry that are shared across independently formed hybrid populations. Genome-wide we observe over a hundred such deserts in *X. birchmanni* × *X. cortezi* populations and find that more than 25% of these minor parent ancestry deserts are repeated across the 2 populations (Fig 4A). Moreover, in cases where deserts are not replicated across populations, minor parent ancestry still tends to be low in the second population (on average falling in the lowest quartile of minor parent ancestry; e.g., Fig 3A). Consistent with our findings that selection acted early after hybridization, we find that minor parent deserts are typically large (on average 1.8 Mb). These regions are exciting candidates to pursue as we begin to map hybrid incompatibilities between *X. birchmanni* and *X. cortezi* in natural populations and in the laboratory [67].

Beyond these genome-wide patterns, we know the precise locations of 2 loci that cause a lethal mitonuclear incompatibility in *X. birchmanni* × *X. cortezi* hybrids when they are mismatched with mitochondrial ancestry [58,69]. If selection on hybrid incompatibilities is responsible for local deviations in ancestry in *X. birchmanni* × *X. cortezi* hybrid populations, we should see biased ancestry in these specific regions of the genome in both hybrid populations. Indeed, we identify large regions depleted of minor parent ancestry surrounding the genes involved in lethal mitonuclear incompatibilities on chromosome 6 (Fig 4C) and chromosome 13 (S20 Fig). We find suggestive evidence of selection at 2 other regions that are involved in $F_2$ inviability but have not been fine mapped (S22 Fig; [67]). Based on these results at known incompatibilities, we infer that shared local ancestry patterns in *X. birchmanni* × *X. cortezi* hybrid populations are at least in part driven by strong selection against hybrid incompatibilities.

We also observed unexpectedly large overlap in regions of the genome where minor parent ancestry is elevated across the 2 populations. Eighty-nine of the 147 regions with elevated minor parent ancestry in Chapulhuacanito were also elevated in the Santa Cruz population (approximately 60%). This enrichment may indicate that *X. birchmanni* ancestry in these regions is beneficial to hybrids, although we found no patterns of gene enrichment within islands that exceeded expectations by chance (Text L in S1 File), nor overlap with previously mapped QTL for sexually selected traits or ecological adaptations in *Xiphophorus* species [70,71]. The combined dynamics of genome-wide selection against deleterious and adaptive variation in hybrids are poorly understood in most cases (but see [11,20,29]), pointing to exciting directions for future work.

The variety of hybrid populations within *Xiphophorus* allowed us to ask how predictability of genome evolution after hybridization varies with genetic divergence. We analyzed replicate hybrid populations formed between both *X. birchmanni* and *X. malinche* and *X. birchmanni* and *X. cortezi*. Since *X. birchmanni* and *X. malinche* are more closely related than *X.*

*birchmanni* and *X. cortezi*, theory predicts that the total strength of selection on *X. birchmanni* × *X. malinche* hybrids across the genome should be weaker [40]. Notably, the correlations in local ancestry we observed in the *X. cortezi* × *X. birchmanni* hybrid populations greatly exceed those observed in *X. birchmanni* × *X. malinche* hybrid populations. Comparisons across hybrid population types (i.e., comparing *X. cortezi* × *X. birchmanni* hybrid populations to *X. birchmanni* × *X. malinche* hybrid populations) yield the lowest predictability in minor parent ancestry (S8 Table). Our wavelet analyses suggests that repeatability across hybrid population types is limited to the broadest genomic scales, potentially reflecting the effects of shared genomic architecture rather than shared targets of selection. This result is consistent with the idea that loci involved in hybrid incompatibilities may arise idiosyncratically between lineages, as different sets of mutations fix along different evolutionary branches. We note that while *X. cortezi* × *X. birchmanni* populations tend to be older than *X. birchmanni* × *X. malinche* populations (S1 Fig; [35]), wavelet analyses suggest that in both cases much of the observed variation in minor parent ancestry along the genome is established in the earliest generations following hybridization (Fig 3; [38]).

Hybridization is a common evolutionary process that profoundly shapes genome evolution, adaptation, and trait evolution. Our accurate local ancestry inference approaches allowed us to uncover striking repeatability in local ancestry across independently formed *X. birchmanni* × *X. cortezi* hybrid populations and begin to unravel the fundamental question of how these patterns scale with evolutionary divergence between species [72]. We find that this strong predictability in genome evolution is impacted both by local factors like the locations of hybrid incompatibilities and global factors such as the recombination landscape in the genome. While the extent to which the patterns observed in *Xiphophorus* hybrids are generalizable to other hybridizing species awaits future investigation, such strong predictability in genome evolution after hybridization has wide-ranging implications for the many species groups across the tree of life in which hybridization is common.

## Methods

### Sample collection

Samples for low-coverage whole genome sequencing were collected from 2 different geographical regions (Fig 1). Wild fish were collected using baited minnow traps in Hidalgo and San Luis Potosí, Mexico. We previously identified hybrids between *X. birchmanni* and *X. cortezi* at multiple sites on the Río Santa Cruz in northern Hidalgo [23,68]. We continued to sample from these sites for the present analysis (Huextetitla—21°9′43.82″N 98°33′27.19″W and Santa Cruz—21°9′27.63″N 98°31′13.79″W). We also added a new site in a different drainage (Fig 1), near the town of Chapulhuacanito (21°12′10.58″N 98°40′28.27″W). This site also contained *X. birchmanni* × *X. cortezi* hybrids (see Results), but this hybridization event is clearly independent given the geographical distance between these locations and other evidence (Text B–D in S1 File). At both collection sites, nearly pure *X. birchmanni* individuals were also sampled. These individuals were identified based on their genome-wide ancestry and excluded from further analysis.

We combined previously collected data sets from the Río Santa Cruz (N = 254; [23,68]) with 216 new samples collected from Chapulhuacanito in June of 2021. Collected fish were anesthetized in 100 mg/ml buffered MS-222 and water, following Stanford APLAC protocol #33071. A small fin clip was taken from the caudal fin of each individual and preserved in 95% ethanol for later DNA extraction.

For this study, we also took advantage of historical collections from 2003, 2006, and 2017 in the same regions. These samples were matched to present-day collection sites using GPS

coordinates and represented a mix of fin clips preserved in DMSO and whole fish preserved in 95% ethanol. We prepared libraries, sequenced all samples, and identified 76 hybrids from historical samples from Chapulhuacanito and 23 from the Río Santa Cruz.

## Chromosome scale assembly for *X. cortezi*

We generated a new reference genome for *X. cortezi* for this project from a lab-raised male descended from an allopatric population sampled on the Río Huichihuayán. Previous work involving *X. cortezi* used a draft genome assembled with 10× chromium linked read technology [23,68]. We assembled the new reference using PacBio HiFi data.

Genomic DNA was isolated from tissue using QIAGEN's Genomic-Tip 500/G columns following the manufacturer's recommendations with some adaptations. Approximately 400 mg of body tissue was digested in 1.5 ml of Proteinase K and 19 ml Buffer G2 at 50˚C for 2 h, inverting the sample every half hour. Following the incubation, the column was equilibrated using 10 ml of Buffer QBT. The sample was vortexed for 10 s at maximum speed, then immediately applied to the column. Two washes were performed with a total of 30 ml of Buffer QC. The column was then transferred to a clean 50 ml tube and genomic DNA was eluted from the column with 15 ml of Buffer QF that was prewarmed to 50˚C. The DNA was precipitated using 10.5 ml of isopropanol, mixed gently, then centrifuged immediately at a speed of $5,000 \times g$ for 15 min at 4˚C. The DNA pellet was then washed with 4 ml of cold 70% ethanol and re-pelleted via centrifugation. Then, the pellet was air-dried for 10 min and resuspended in 1.5 m of Buffer EB. Genomic DNA was quantified and assessed for quality using a Qubit fluorometer, Nanodrop, and Agilent 4150 TapeStation. Extracted DNA was sent to Admera Health Services, South Plainfield, NJ for PacBio library prep and sequencing on SMRT cells. Raw sequence data is available on NCBI's Sequence Read Archive (BioProject PRJNA1106506).

To remove residual adapter contamination from the HiFi reads, we used HiFiAdapterFilt [73] with the default match parameter of 97% and a length parameter of 30 bp. We then generated a phased genome assembly with hifiasm (v0.16.1; [74]). The resulting primary assembly was 144 contigs with a N50 of 28,997,520 bp. To achieve a chromosome-level assembly, we scaffolded the *X. cortezi* genome to the chromosome-level genomes of species in its sister clade: *X. birchmanni* and *X. malinche* (NCBI assembly id: GCA_036418095.1) using RagTag (v2.1.0; [75]). Where these scaffolded genomes differed in synteny, we used the chromosome-level assemblies of *X. hellerii*, *X. maculatus*, and *X. couchianus* as outgroups to select the ancestral orientation for *X. cortezi*. This scaffolded *X. cortezi* genome had a scaffold N50 of 32,220,398 bp and length of 723,632,656 bp. These putative *X. cortezi* chromosomes were aligned to the *X. maculatus* genome assembly using minimap2 (v2.24; [76]) and oriented and numbered according to identity with *X. maculatus*.

Chromosome 21 is known to contain the major sex determination locus in many *Xiphophorus* species [77]. To resolve potential structural variation at this locus and include both X and Y linked sequence in the *X. cortezi* reference genome, we generated an alignment between the 2 inferred haplotypes for chromosome 21. We found that one chromosome 21 haplotype was syntenic to chromosome 21 in *X. birchmanni*, while the other contained a 7 Mb chromosomal inversion relative to *X. birchmanni*, which is syntenic to all other *Xiphophorus* species and likely represents the ancestral *Xiphophorus* arrangement of the Y-chromosome [78].

The mitochondrial genome was assembled from the adapter-filtered HiFi reads using Mito-HiFi (v3.2; [79]) with default parameters and using the *X. maculatus* mitochondrial genome as a reference. We used BLASTn [80] searches to identify and subsequently remove mitochondrial contaminant sequences present in the nuclear genome, which were present on only 6 contigs that were all less than 40 kb in length. Following contaminant removal, the

mitochondrial genome assembled with MitoHiFi was added to the *X. cortezi* assembly. The final assembly is available on NCBI (Bioproject PRJNA1106506).

## Annotation of the *X. cortezi* assembly

The *X. cortezi* genome was annotated using a pipeline adapted from a previous study [81]. Transposable elements (TEs) in the assembly were identified using RepeatModeler and Repeat-Masker [82]. RepeatModeler was first used for an automated genomic discovery of TE families in the assembly. This result, together with Repbase and FishTEDB [83,84], was input into RepeatMasker for an additional retrieval of TEs based on sequence similarity. For protein coding gene annotation, TEs from known families were hard-masked and simple repeats were soft-masked from the assembly. We used a tool designed to parse RepeatMasker output files [85] to compute quantitative information on representation of different TE families. We repeated this approach for the *X. birchmanni* PacBio reference assembly generated using the same approach. Analysis of differences between the 2 species in repeat content is available in Text A in S1 File.

Protein coding genes were annotated by collecting and synthesizing gene evidence from homologous alignment, transcriptome mapping, and ab initio prediction. For homologous alignment, 455,817 protein sequences were collected from the vertebrate database of Swiss-Prot (https://www.uniprot.org/statistics/Swiss-Prot), RefSeq database (proteins with ID starting with "NP" from "vertebrate_other"), and the NCBI genome annotation of human (GCF_000001405.39_GRCh38), zebrafish (GCF_000002035.6), platyfish (GCF_002775205.1), medaka (GCF_002234675.1), mummichog (GCF_011125445.2), turquoise killifish (GCF_001465895.1), and guppy (GCF_000633615.1). We then aligned those protein sequences onto the assembly using both GeneWise and Exonerate (https://www.ebi.ac.uk/about/vertebrate-genomics/software/exonerate) to collect homologous gene models. In order to speed up GeneWise, GenblastA was used to retrieve the rough alignment region of the assembly for each protein [86].

For transcriptome mapping, we used previously collected RNA-seq reads from multiple tissues [17], cleaned them using fastp [87], and mapped them to the assembly using HISAT [88]. StringTie was then used to interpret gene models from the mapping results [88]. In parallel, we used Trinity to assemble RNA-seq reads into transcript sequences and aligned them to assembly for gene modeling using Splign [89,90].

We used AUGUSTUS for the ab intio gene prediction [91]. AUGUSTUS was trained for the first round using BUSCO genes. Genes that were predicted repeatedly by Exonerate, Gene-wise, StringTie, and Splign were considered to be high-quality genes and were used to train AUGUSTUS for the second round. All collected homologous and transcriptome gene evidence were used as hints for AUGUSTUS for the ab initio gene prediction.

To generate the final consensus annotation, we screened homology gene models locus by locus. When 2 gene models competed for a splice side, we kept the one better supported by transcriptome evidence (using transcriptome data from [17]). When a terminal exon (with a start/stop codon) from an ab initio or homology gene model was better supported by transcriptome data than that of the previously selected gene model, the exons in question were replaced by the predictions of the gene model best supported in the transcriptome data. We also kept an ab initio prediction when its transcriptome support was 100% and it had no homology prediction competing for splice sites.

## Low coverage whole genome sequencing

We extracted DNA from fin clips collected from wild-caught fish using the Agencourt DNAdvance kit (Beckman Coulter, Brea, California, United States of America). We used half-

reactions but otherwise followed the manufacturer's instructions for DNA extraction. We used a BioTek Synergy H1 (Agilent, Santa Clara, California, USA) microplate reader to quantify extracted DNA. We diluted DNA to a concentration of 10 ng/µl and then prepared tagmentation-based libraries from this genomic DNA for low-coverage whole genome sequencing. The approach used for generating libraries is described in Langdon and colleagues [23]. Dual-indexed libraries were bead purified with 18% SPRI magnetic beads, quantified on a qubit fluorometer (Thermo Scientific, Wilmington, Delaware, USA), and visualized on an Agilent 4200 Tapestation (Agilent, Santa Clara, California, USA). Purified libraries were sequenced by Admera Health Services (South Plainfield, New Jersey, USA) on an Illumina HiSeq 4000 instrument.

## Whole genome resequencing

To evaluate patterns of genetic variation within ancestry tracts, we sequenced a subset of individuals ($N$ = 3 per genotype per population) at high coverage. For these individuals, we prepared libraries following the approach of Quail and colleagues [92]. We used 500 ng to 1 µg of DNA per sample and sheared this input DNA to approximately 400 bp using a QSonica sonicator. The fragmented DNA underwent an end-repair reaction with dNTPs, T4 DNA polymerase, Klenow DNA polymerase, and T4 PNK for 30 min at room temperature. An A-tail was added to the end-repaired DNA using a mix of Klenow exonuclease and dATP, incubated for 30 min at 37˚C. The A-tail facilitated ligation of adapters with DNA ligase in a 15-min reaction performed at room temperature. The resulting sample was purified using the Qiagen QIAquick PCR purification kit. Barcodes were added during a final PCR amplification step using the Phusion PCR kit, which was run for 12 cycles. This reaction was purified with 18% SPRI beads and libraries were visualized on the Agilent 4200 Tapestation and quantified using a Qubit fluorometer. These libraries were also sent to Admera Health Services for sequencing on an Illumina HiSeq 4000 machine.

## Inferring recombination maps for *X. birchmanni* and *X. cortezi*

In past work, we used population genetic methods to infer a linkage disequilibrium (LD)-based recombination map for an earlier version of the genome assembly for *X. birchmanni* [35]. We repeated the same approaches with the new *X. birchmanni* reference genome to generate a new LD-based map. Briefly, we used the previously published resequencing data for 22 adult *X. birchmanni* individuals and a pedigreed family with 5 offspring [35], for a total of 24 unrelated adults. We mapped reads to the genome with *bwa* mem, realigned indels with PicardTools, and called variants with GATK (v3.4; [93]). We filtered variant and invariant sites based on quality thresholds as we had with the original recombination map (DP < 10; RGQ < 20; QD < 10; MQ < 40; FS > 10; SOR > 4; ReadPosRankSum< -8; MQRankSum < -12.5). We excluded sites that overlapped with annotated repetitive regions or had <0.5× or >2× the average genome-wide coverage for that individual. For invariant sites, only RGQ and DP filters could be used. Using this filtered list of sites, we inferred the expected error rate with plink [94] using expectations of mendelian segregation in the pedigree. Finding evidence of a low error rate (approximately 0.45% per SNP across 5 offspring), we first removed these errors and then proceeded to phasing and inferring the LD map. We performed phasing using the program shapeit2 with the duohmm flag for inclusion of family data [95]. Past simulations matching parameters observed in *X. birchmanni* have suggested that although phasing likely introduces errors, improvements in map resolution outweigh errors introduced by phasing [35].

We inferred the LD map using LDhelmet. LDhelmet relies on a mutation transition matrix for recombination map inference [96] and also can take advantage of distributions of ancestral

alleles when computing likelihoods. To infer ancestral alleles for both purposes, we used phylo-fit [97]. Previous simulations matching parameters observed in *X. birchmanni* have suggested that this approach results in accurate inference of ancestral sequences [35]. We used previously collected whole genome sequence data from 11 species of *Xiphophorus* (S13 Table) to infer the likely ancestral basepair at variable sites as described previously [35] using the prequel command [97]. To run phylofit, we provided the aligned sequences and the inferred species tree for these groups of species [98]. For mutation matrix inference based on phylofit output, we used a threshold of 0.99 to convert posterior probabilities for the ancestral basepair to hard calls.

We then used phased haplotypes from all unrelated *X. birchmanni* individuals (48 haplotypes in total) and the mutation transition matrix to infer an LD-based recombination map with LDhelmet [96]. The total number of SNPs input into LDhelmet was 2,565,331. We first computed a likelihood lookup table for ρ values using a grid table ranging from 0 to 10 (sampling in intervals of 0.01 from 0–1 and 1 from 1–10). We next inferred recombination rates using LDhelmet's rjMCMC procedure with a block penalty of 50, a burn-in of 100,000, and ran the Markov chain for 1,000,000 iterations. Past work has suggested that a block penalty of 50 improves accuracy for inference of broad scale recombination rates in *Xiphophorus* [35]. Following map inference, we excluded SNP intervals with implausible high recombination rates ($\rho$/bp $\geq 0.4$) and summarized recombination rates in windows of physical distance ranging from 5 kb to 5 Mb. We also used the local recombination rate estimates and the inferred lengths of each chromosome in cMs to divide the chromosome into windows of genetic distance for certain analyses (Text N in S1 File).

Because we had access to whole genome resequencing data for 9 unrelated *X. cortezi* individuals from the Huichihuayán river (near the Nacimiento) from previous work [17,23], we decided to supplement this data to build an LD-based map for this species as well. To generate a comparable sample size for this inference, we sequenced an additional 8 individuals following the whole genome resequencing protocol described above. The average coverage for the *X. cortezi* individuals was approximately 65×, and the range was 19–113×. We inferred an LD-based map for *X. cortezi* as described above, except that we lacked access to pedigree data for mendelian error correction and phasing.

With the lower sample size and lack of pedigree data, we expected the *X. cortezi* map to be less accurate than the *X. birchmanni* map but used it to test general hypotheses. Swordtails have deleted the N-terminal domain of PRDM9 [99] and have a conserved PRDM9 zinc-finger binding domain across the clade. Past work has indicated that swordtails behave as PRDM9 knock-outs with a higher frequency of recombination events near the TSS, CpG islands, and H3K4me3 marks [35,99]. Using the inferred LD map for *X. cortezi*, we confirmed that we observe elevated recombination rates close to the TSS and H3K4me3 peaks, similar to patterns observed in *X. birchmanni* (S10 Fig; Text F in S1 File). We note that the median inferred ρ/bp in *X. cortezi* was substantially higher than in *X. birchmanni* (0.0027 versus 0.00076). Based on the results of our analyses of historical population sizes (see Text F in S1 File), we expect to see elevated ρ/bp in *X. cortezi* since ρ reflects $4Ne^*r$ and we infer that *X. cortezi* has had approximately 2× the effective population size of *X. birchmanni* over the past 100k generations. However, ρ/bp may also be impacted by a higher error rate in the *X. cortezi* recombination map given the lack of access to pedigree data.

## Changes to the local ancestry inference pipeline

We previously developed approaches for local ancestry inference for hybrids between *X. birchmanni* and *X. cortezi*, but we made several improvements upon previous implementations for

this project. First, we used a new chromosome scale assembly for *X. cortezi* generated by PacBio HiFi technology (see above). To more accurately quantify allele frequencies in the parental species, we sampled additional allopatric populations of *X. cortezi*, which has been less intensively sampled from a genomic perspective than *X. birchmanni*, and sequenced F$_1$ hybrids between the 2 species for error correction. We also identified and corrected an error in the *ancestryinfer* code (https://github.com/Schumerlab/ancestryinfer) that had resulted in a number of ancestry informative sites being erroneously excluded in previous versions of the pipeline.

Using the new assemblies, we identified candidate ancestry informative sites by aligning resequencing data from a high-coverage *X. cortezi* individual to the *X. birchmanni* PacBio assembly (as described previously; [17,57,100]), and identifying all sites that were homozygous for different states in this data. We then treated these sites (2.64 million) as potential ancestry informative sites and evaluated their frequency in allopatric *X. cortezi* and *X. birchmanni* populations using 1× whole genome sequence data from 90 individuals of each species from 3 source populations each (*X. birchmanni*: Coacuilco, Talol, Xaltipa; *X. cortezi*: Puente de Huichihuayán, Octzén, Calle Texacal). We identified sites that had a 98% or greater frequency difference between the 2 species at our filtered set of ancestry informative sites (1,001,684 sites). We used these sites and their observed frequencies in the parental species as input for our ancestry HMM pipeline (*ancestryinfer*; [100]).

We next took advantage of our lab-generated F$_1$ hybrids to further filter these ancestry informative sites. We collected approximately 1× whole genome sequence data for 42 F$_1$ hybrids we generated between *X. birchmanni* and *X. cortezi* and analyzed these individuals using the *ancestryinfer* pipeline, specifying the *X. birchmanni* reference as genome 1 and the *X. cortezi* reference as genome 2. We set the error rate to 0.02 for this initial analysis. After running the pipeline, we converted posterior probabilities for each ancestry state into hard-calls using a posterior probability threshold of 0.9. Because F$_1$ hybrids should be heterozygous for all ancestry informative sites across the genome, we identified ancestry informative sites that were called with high confidence as homozygous *X. birchmanni* or homozygous *X. cortezi* and excluded these sites. This resulted in a final set of 995,825 ancestry informative sites which we used for downstream analyses, or a median of one marker every 240 basepairs across the 24 major chromosomes.

We tested the performance of this approach on 30 *X. cortezi* individuals we had not used in our initial filtering, 12 *X. birchmanni* individuals, 13 F$_1$ hybrids, 26 F$_2$ hybrids, and 5 BC$_1$ hybrids (backcrossed to *X. cortezi*) where we have clear expectations for true ancestry. Based on this analysis, we found that performance of the HMM approach was excellent (S13–S15 Figs).

### Local ancestry inference and processing for downstream analysis

Using the ancestry informative sites described above, we next proceeded to analyze hybrid individuals from Chapulhuacanito and the Río Santa Cruz using the *ancestryinfer* pipeline. We inferred local ancestry for 291 individuals from Chapulhuacanito and 277 individuals from the Río Santa Cruz. Because previous analyses have indicated that *ancestryinfer* is not sensitive to priors for initial admixture time and admixture proportions [100], we set the prior for the genome-wide admixture proportion to 0.5 and the prior for the number of generations since initial admixture to 50. However, we repeated local ancestry inference for both populations following demographic inference using ABCreg (see next section) using priors inferred from this analysis for initial admixture time and admixture proportion. We found that our results were qualitatively unchanged (S10 Table). For all analyses, we used a uniform recombination prior, set to the median per-basepair recombination rate in Morgans inferred for *X. birchmanni*.

For a number of downstream analyses, it was useful to convert posterior probabilities for different ancestry states into hard-calls. As we have previously, we used a posterior probability threshold of 0.9 or greater to assign an ancestry informative site to a given ancestry state (e.g., homozygous *X. birchmanni*, heterozygous for ancestry, or homozygous *X. cortezi*). Ancestry informative sites with lower than a 0.9 probability for any ancestry state were masked. We also filtered out sites that were covered in fewer than 25% of individuals. This resulted in 994,891 sites across the genome in Santa Cruz and 994,906 sites across the genome in Chapulhuacanito for downstream analysis. All local ancestry results are available on Dryad (DOI: 10.5061/dryad.qnk98sfq1).

Consistent with previous work [57,68], a subset of the individuals we sequenced were nearly pure *X. birchmanni* (>98% of the genome derived from the *X. birchmanni* parent species). We identified and excluded these individuals from our data set before examining patterns of local ancestry within the 2 hybrid populations, resulting in a data set of 114 hybrid individuals from Chapulhuacanito and 276 from the Río Santa Cruz populations. We summarized minor parent ancestry across individuals by average ancestry hard-calls in non-overlapping windows of a range of sizes (e.g., 100 kb to 500 kb and 0.1 to 0.5 cM).

## Demographic inference in the Chapulhuacanito and Santa Cruz populations

To inform our understanding of patterns of local ancestry along the genome in Chapulhuacanito and Santa Cruz, we wanted to better understand the likely demographic history of these populations. To do so, we used a regression-based Approximate Bayesian Computation or ABC approach with the software ABCreg [62]. We previously applied a similar approach to infer the likely demographic history of the Santa Cruz population [23] but repeat it for both populations here taking advantage of our larger empirical data sets and updated local ancestry inference pipeline. All simulations were performed in SLiM [61].

For each simulation, we drew each population demographic parameter from a uniform or log-uniform prior distribution, performed simulations, and calculated summary statistics for the simulation. We recorded the summary statistics and simulated parameters and compared them to the same statistics calculated from the real data. We modeled one chromosome 25 Mb in length with local recombination rates matching those observed on *X. birchmanni* chromosome 2. We used both global and local metrics as summary statistics (S14 Table). We used the tree sequence recording functionality of SLiM to determine local ancestry of each individual in the hybrid population [101]. To perform each simulation, we used the following steps:

1. We initialized parental populations and formed a hybrid population between them. We determined the admixture proportion by drawing from a uniform prior distribution for the proportion of the genome derived from parent 1 (0.5–1). Similarly, we determined the hybrid population size for the simulation by drawing from a uniform prior ranging from 2–10,000 individuals.

2. We drew a migration rate from each parental species from a log uniform prior distribution (ranging from $m = 0\%$–3% per generation based on previous results [101]).

3. We drew a time since initial admixture parameter from a uniform distribution ranging from 10 to 400 generations.

4. We performed the simulation for the number of generations drawn in step 3 above, implementing migration from the parental species each generation at the rate drawn in step 2.

5. We randomly sampled 69 and 242 individuals from the population to match the number of hybrid individuals sampled in Chapulhuacanito and Santa Cruz in 2021 and 2020, respectively.

6. We generated summary statistics to compare to summary statistics calculated based on the real data.

7. We repeated this procedure until 150,000 simulations had been generated.

8. We ran the program ABCreg with the tolerance parameter set to 0.005.

To evaluate the validity of our approach, we tested how well this procedure worked to infer parameters for a simulated population with known history. We randomly sampled 100 simulations generated as described above and treated these simulations as if they were the real data. We calculated summary statistics and ran ABCreg as described above (excluding these simulations from the full ABCreg data set). We then calculated the 95% quantile of the posterior distribution for each demographic parameter and asked how well this distribution captured the known parameters for the focal simulation. In general, the 95% quantile of the posterior distributions for each test set overlapped with the true value (S25 Fig). However, performance was poorer when we asked how often the true value fell in the 50% quantile of the posterior distribution produced by ABCreg, indicating that we should view MAP estimates as approximate estimates of the likely demographic history of each population (S25 Fig).

For all subsequent simulations modeling population evolution in Chapulhuacanito and Santa Cruz, we based demographic parameters on the results of these simulations. Text G in S1 File for more information.

## Repeatable patterns of minor parent ancestry as a function of genomic architecture

Using the LD-based recombination map described above, we evaluated evidence for a correlation between the local recombination rate in windows and minor parent ancestry in those same windows. As previously reported [57], we found strong correlations between local recombination rate and average minor parent ancestry in the Santa Cruz population, with minor parent ancestry being more common in regions of the genome with the highest local recombination rates. We repeated this analysis for the Chapulhuacanito population and replicated this pattern.

Because recombination events in *Xiphophorus* species appear to disproportionately localize to functionally dense regions of the genome (e.g., transcriptional start sites, CpG islands, and H3K4me3 peaks; [24,99]), we wanted to control for proximity to some of these elements in our analyses. We calculated the number of coding and conserved basepairs in each window and incorporated this into our analysis. We calculated the Spearman's partial correlation between recombination rate, minor parent ancestry, and coding (or conserved basepairs) across a range of window sizes (S6 Table). We also repeated this analysis calculating average ancestry and the number of coding (or conserved) basepairs in windows of a particular genetic length (0.1 to 0.5 cM; S7 Table; Text N in S1 File).

## Cross-population repeatability in ancestry

The Santa Cruz and Chapulhuacanito populations occur in separate river systems and multiple lines of evidence indicate that they originated from independent hybridization events between *X. birchmanni* and *X. cortezi*. We wanted to understand the extent to which local ancestry between these 2 populations was correlated. Presumably, correlations that are observed (barring those due to technical artefacts) should be driven by shared sources of selection, either due to shared loci under selection or shared genomic architecture (e.g., similar recombination maps and locations of coding and conserved basepairs between species).

To evaluate this, we used a Spearman's correlation test implemented in R. We calculated these correlations in windows of a range of physical sizes (100 kb to 500 kb) and genetic sizes (0.1 to 0.5 cM). We performed each of these calculations thinning the data to retain only a single window every Mb or a single window every 1.5 cM (typically approximately 600 kb in *Xiphophorus*). This analysis should be conservative since admixture LD decays to background levels over ~500 kb in Santa Cruz and Chapulhuacanito (S11 Fig). We found that the cross-population correlations in local ancestry in these analyses were surprisingly high (S8 Table). Given this observation, we sought to exclude several technical factors that might be artificially inflating this correlation.

Because power to infer ancestry will vary along the genome, we wanted to evaluate whether accounting for this power variation impacted the signal we observed. Certain regions of the genome have a higher density of ancestry informative sites between *X. birchmanni* and *X. cortezi*. We determined the median distance between ancestry informative sites (240 bp), and thinned markers such that in regions with higher marker frequency, we retained at most 1 marker per 240 bp, and repeated local ancestry inference with this marker set. We also identified and excluded windows in which we have especially low power to infer ancestry (the number of ancestry informative sites fell in the lower 5% quantile of the genome-wide distribution).

We investigated the impact of removing other regions of the genome where we might expect to have a higher error rate in local ancestry inference. Analysis of our assemblies using seqtk telo [102] suggest that some of our chromosomes include assembled telomeric regions (S19 Fig). Since these regions may be especially challenging to analyze, we recalculated cross-population correlations excluding any region within 1 Mb of the end of a chromosome. We also generated a version of the ancestry informative sites excluding markers that overlapped with repetitive regions and recalculated cross-population correlations. We performed a number of other complementary analyses that are described in detail in Text G–I in S1 File.

Overall, our qualitative results were unchanged in each of the modifications described above and in the series of additional analyses described in Text G–I in S1 File (S10 Table). As a sanity check, we also performed analyses where we generated subpopulations from either Santa Cruz or Chapulhuacanito and asked about the observed correlations in ancestry when individuals truly originate from the same population. We also compared correlations in ancestry between samples from the same population over time, and from different sites in the same river. Reassuringly, all of these comparisons yielded correlations in ancestry that exceeded what we observed between independently formed populations at Santa Cruz and Chapulhuacanito (S8 Table and S12 Fig).

## Evidence of independent formation of Santa Cruz and Chapulhuacanito

While the Río Santa Cruz and Río San Pedro are separated by over 130 km of river miles, we wanted to perform additional analyses to confirm that they were independent in origin given the strong correlations in local ancestry that we observe across the 2 populations. To do so, we used a combination of approaches. First, we performed principal component analysis of the locations of observed ancestry transitions in Santa Cruz and Chapulhuacanito. If these populations formed and evolved independently, we would expect that observed ancestry transitions (which reflect recombination events in the ancestors of each hybrid individual) would occur largely in different locations across the 2 populations.

To generate a data set for principal component analysis, we first identified the approximate locations of ancestry transitions for each hybrid individual in our data sets from Santa Cruz and Chapulhuacanito, defined as the interval over which the posterior probability changes

from 0.9 posterior probability for one ancestry state to 0.9 for another ancestry state. Ancestry transitions that were supported by flanking segments of <5 kb were removed. Qualitatively, this removed transitions where the ancestry state switched and then immediately reverted, which we hypothesized were likely to be errors. Once ancestry transitions have been identified, we binned the genome into windows of 0.5 cM, and for a given individual recorded a 1 if there was a transition observed in that window and a zero if there was not. While most ancestry transitions were well resolved (mean 13.6 kb), some spanned multiple windows, so we used the midpoint as the location of the ancestry transition for these purposes. We ran a principal component analysis of this matrix in R.

While the variation in ancestry transition locations suggested broadly different histories of recombination in the Santa Cruz and Chapulhuacanito populations, this pattern could also be consistent with an initial period of shared history followed by vicariance. Thus, as a complementary approach, we quantified how frequently the locations of ancestry transitions were shared between pairs of individuals in the Santa Cruz and Chapulhuacanito populations, compared to expectations by chance. Shared ancestry transitions, reflecting ancestral recombination events, could occur by chance due to recombination hotspots or poor resolution of the precise locations of recombination events, but an excess of shared transitions is likely to reflect shared ancestors and thus a shared population history. For both the real and simulated data, we excluded ancestry transitions that were poorly resolved (>250 kb in length) from our analysis. For the real data, we quantified how frequently the intervals of ancestry transitions overlapped in pairs of individuals from the 2 populations using bedtools [103], treating ≥1 basepair shared as an overlap. To generate simulated data, we performed a series of steps. We first excluded windows where ancestry for 1 parental species had fixed in the real data, as these regions of the genome cannot contain ancestry transitions in the real data. For each individual, we iterated through all of the ancestry transitions observed, randomly sampling a new location for each ancestry transition, weighted by the *X. birchmanni* recombination map (summarized in 100 kb windows). Within that randomly selected window, we used R's runif function to identify a start position of the recombination interval and set the stop position based on the interval length. We repeated this until all ancestry transitions for an individual had been assigned a random position weighted by the recombination map and repeated this process for all individuals. Next, we quantified the overlap of ancestry transitions in pairs of simulated Santa Cruz and Chapulhuacanito individuals relative to the real data. These results are shown in Fig 1D.

To further exclude the possibility that observed patterns might be generated by an initial period of shared history followed by vicariance, we performed several additional analyses. We collected high-coverage whole genome sequencing data (>20×) for 3 hybrid individuals from each population and 3 pure *X. birchmanni* individuals found in the same populations (i.e., individuals inferred to derive >98% of their genome from *X. birchmanni*). For these individuals, we called variants throughout the genome as described above for inference of LD-based recombination maps (mapping and variant calling were performed using both references independently, and results were qualitatively similar). Using this variant information, we performed a principal component analysis on the observed variants genome wide in the individuals collected from Santa Cruz and Chapulhuacanito as well as previously collected high-coverage data from source populations of *X. birchmanni* (Coacuilco; [35]) and *X. cortezi* (Huichihuayán and Puente de Huichihuayán; [57,68]).

Because hybrids combine the genomes of the *X. birchmanni* and *X. cortezi* individuals that contributed to the hybridization event, we were also interested in subsetting these regions of the genome and analyzing them separately. To do so, we conducted local ancestry inference on the 6 hybrid individuals as described above, and identified regions where we had high

confidence that all 6 hybrid individuals in our data set were homozygous *X. cortezi* or homozygous *X. birchmanni*. We extracted the variants in these segments from hybrids and from the corresponding parental species plink files (i.e., from *X. cortezi* individuals for analysis of homozygous *X. cortezi* ancestry tracts). We performed a separate PCA on the *X. cortezi* and *X. birchmanni*-derived regions in hybrids. If Santa Cruz and Chapulhuacanito somehow shared a population history, we would expect these regions to cluster closely together and potentially overlap in a principal component analysis. See Text E in S1 File for a more detailed discussion of these results and their implications.

To more closely evaluate possible relatedness in these ancestry tracts, we used the program GCTA to generate a genetic relatedness matrix for these 6 hybrid individuals [65]. We performed separate analyses for the *X. cortezi* and *X. birchmanni* ancestry tracts. As above, we only analyzed regions where all 6 hybrid individuals were homozygous *X. cortezi* or *X. birchmanni* in that region, respectively. We included data from the relevant parental populations for comparison. We found that all hybrid individuals from the same population were inferred to have some degree of relatedness based on analysis of both *X. cortezi* and *X. birchmanni* derived ancestry tracts, but all values for cross-population comparisons were negative (S6 Fig).

We also used our high-coverage data from the parental and hybrid populations to infer evidence for IBD tracts using the program IBDseq [66]. Because we do not have access to a large number of high-coverage individuals, we were unsure whether programs designed for large data sets like IBDseq would have reasonable performance on our data. We thus first analyzed a known *X. birchmanni* pedigree [35] with IBDseq using the parameters errormax = 0.0025, r2window = 100, and r2-max = 0.5 (we use a higher than recommended r2-max due to high admixture LD in our focal populations). We used an LOD threshold of 3 as a cutoff for reporting an IBD tract [66]. When applied to the *X. birchmanni* pedigree data, we found that IBDseq had excellent performance. We thus applied the same approach to identify IBD tracts in the 6 high-coverage individuals from Chapulhuacanito and Santa Cruz, and in different iterations of the analysis also included high-coverage individuals from allopatric and sympatric parental populations (see Text D in S1 File). However, we note that with our small sample size we might expect more variable performance in IBD tract identification in a sample of less closely related individuals.

Since most of the genome of individuals in the 2 hybrid populations is derived from *X. cortezi*, we performed an additional analysis focusing on *X. cortezi* ancestry tracts in the high-coverage individuals. Reasoning that populations derived from distinct source populations and with independent demographic histories should harbor distinct frequencies of genetic variants, we performed a "mismatch" analysis. We subset our data to focus only on regions that were homozygous *X. cortezi* across all 6 high-coverage hybrid individuals. For each pair of individuals in our data set, we counted each site where individual 1 was homozygous for one allele and individual 2 was homozygous for another (within *X. cortezi* ancestry tracts). We counted the total number of these sites along the genome, divided by the total number of sites that passed quality thresholds in both individuals (within *X. cortezi* ancestry tracts), and treated this as our mismatch statistic. We compared this mismatch statistic within-populations versus between-populations (Fig 1G) and to population genetic simulations in SLiM modeling distinct population histories (Text C in S1 File).

## Spatial scale of cross-population correlations in ancestry

In the absence of selection or genetic drift, the ancestry proportion in a hybrid population would remain uniform along the chromosome. In real populations, ancestry varies along the genome due to the combined effects of recombination with genetic drift, selection, and repeated admixture events. The spatial scale of ancestry variation along the genome holds

important information about the timing of demographic and selective events, since recombination progressively shortens ancestry tracts across generations. We took advantage of the recent application of the Discrete Wavelet Transform to decompose correlations between genomic signals into independent components associated with different spatial genomic scales [38]. Briefly, the method transforms a signal measured along the genome (e.g., ancestry) into a set of coefficients that measure changes in the signal between adjacent windows at different locations and with windows of different sizes. The wavelet transform is performed on the 2 signals separately, and the correlation between the coefficients at a given scale for the 2 signals are weighted by the variance at that scale (also determined from the wavelet coefficients) to give the contribution of each scale to the overall correlation. This approach offers the advantage that correlations across scales carry independent information, in contrast to traditional window-based analyses used elsewhere in the manuscript where results across different window sizes are confounded due to the nestedness of windows of different sizes.

As this analysis requires evenly spaced measurements along a chromosome, we first interpolated admixture proportions within diploid individuals to a 1 kb grid for each chromosome, then averaged across individuals to obtain the interpolated sample admixture proportion. We used the inferred recombination maps described above to obtain estimates of recombination in windows centered on the interpolated ancestry measurements. We applied a threshold to recombination values of $\rho \geq 0.005$ (corresponding to 4% of the genome) which we found improved the strength of correlation between genetic lengths of chromosome inferred from the LD map versus from an $F_2$ linkage map.

We used the R package *gnomwav* [38] to estimate wavelet correlations between signals (minor parent ancestry between populations, recombination versus minor parent ancestry) at a series of genomic scales for each, with the smallest scale being the resolution of interpolation, and the largest scale corresponding to variation in signals occurring over roughly half of a chromosome. Wavelet correlations were averaged across chromosomes and error bars were obtained from a weighted jackknife procedure following [38]. To obtain the contribution of each scale to the overall correlation, we weighted correlations by the wavelet variances as described in [38]. We ran these analyses for interpolation distances of 1 kb and 32 kb.

## Shared minor parent deserts and islands

We were interested in identifying regions that were likely under selection in both *X. cortezi × X. birchmanni* hybrid populations. Guided by the results of simulations, we used an ad hoc approach to identify regions with shared patterns of unusual ancestry across the 2 populations (see Text K in S1 File). We first identified ancestry informative sites where the minor or major parent ancestry fell in the lower 5% tail of genome-wide ancestry. We then selected the 0.05 cM window that overlapped this ancestry informative site and confirmed that the broader region fell within the lower 10% tail of genome-wide ancestry. We expanded out in the 5′ and 3′ directions in windows of 0.05 cM from this focal window until we reached a window on each edge that exceeded the 10% ancestry quantile. We treated this interval as an estimate of the boundary of the minor parent ancestry desert or island.

Because this approach may prematurely truncate ancestry deserts and islands (particularly in scenarios with error), in a separate analysis we merged any of deserts (or islands) that fell within 50 kb of each other. We filtered these merged regions to remove any regions with fewer than 10 ancestry informative sites, with fewer than 10 single nucleotide polymorphisms present in the recombination map, or that were less that 10 kb in length.

By defining deserts and islands in 0.05 cM windows, we could easily overlap these regions between different populations and determine how many are shared between sampling sites.

This allowed us to define regions that have shared ancestry patterns between Chapulhuacanito and Santa Cruz despite their independent origin. To compare the observed number of shared ancestry deserts and islands to what we would expect by chance, given the overall patterns of ancestry variation along the genome in the 2 populations, we permuted the data in 0.05 cM windows and asked how frequently ancestry deserts and islands were identified as being shared in *X. birchmanni* × *X. cortezi* populations, as we had with the real data. We repeated this procedure 1,000 times. Based on these permutations, we found that few shared minor parent ancestry deserts or islands were expected by chance (Fig 4A).

Since ancestry in a given window is strongly correlated with ancestry in the neighboring windows, especially at smaller spatial scales, we also wanted to performed permutations that preserved this ancestry structure. Specifically, for Chapulhuacanito, we shifted the window labels of ancestry summarized in 0.05 cM windows by 12.5 cMs, and we asked whether any windows that were major (or minor) parent ancestry outliers in the shifted data overlapped with the ancestry deserts (or islands) identified in Santa Cruz (using the same criteria as in the real data). We repeated this procedure 132 times to fully tile the whole genome. Consistent with the naïve permutation approach described above, we found that few minor parent ancestry outliers in *X. birchmanni* × *X. cortezi* hybrid populations overlapped minor ancestry outliers identified in the other population by chance (S26 Fig).

We were interested in whether any of the shared deserts or islands between Chapulhuacanito and Santa Cruz were also ancestry outliers in *X. birchmanni* × *X. malinche* populations. Given the complexity of implementing simulations that would preserve LD structure across the 5 hybrid populations we wanted to evaluate, we simply performed the naïve simulations in 0.05 cM windows described above. Based on these permutations, we found that few minor parent ancestry outliers in *X. birchmanni* × *X. malinche* hybrid populations are expected to overlap minor parent deserts (or islands) identified in *X. birchmanni* × *X. cortezi* hybrid populations by chance.

### Time series analysis

We were interested in understanding how ancestry at minor parent deserts and islands has changed over time. We focus this analysis on Chapulhuacanito due to insufficient sampling over time from the Río Santa Cruz (both in terms of numbers of hybrids sampled and number of sampling years available). The first samples we have access to from Chapulhuacanito are from 2003, approximately 40 generations ago. However, based on our demographic inference, this population likely underwent approximately 100 generations of evolution between initial hybridization and our first sampling year, meaning that even in our earliest samples we are evaluating ancestry in a late-stage hybrid population.

We focused our analysis on deserts and islands identified in 2021, but our results were qualitatively similar when ascertainment was performed in other years. Using the coordinates determined in 2021, we calculated average minor parent ancestry in the same region in each of the other years sampled (2002, 2006, and 2017). For minor parent islands, which showed greater levels of fluctuation in ancestry over time (see Results), we used a linear model implemented in R to test for a significant relationship between year and minor parent ancestry.

### Ethics statement

Field sample collections were approved by Conapesca Permiso de Pesca de Fomento, permit No. PPF/DGOPA-002/19. Vertebrate animal sampling and experiments were approved by Stanford's Administrative Panel on Laboratory Animal Care under protocol number 33071.

## Supporting information

**S1 File. Supporting Information Methods and Results Texts A–N.**
(DOCX)

**S1 Table. Transposable element classes annotated in *X. birchmanni* and *X. cortezi* with more than a 10% copy number difference in the PacBio HiFi genome assemblies of the 2 species.**
(DOCX)

**S2 Table. Inversions that differentiate *X. birchmanni* and *X. cortezi* based on our PacBio HiFi assemblies.** Genome assemblies were aligned with minimap2 and all inversions greater than 100 kb were annotated.
(DOCX)

**S3 Table. MAP estimate and 95% percent confidence intervals for posterior distributions generated by ABCreg for each demographic parameter of interest.**
(DOCX)

**S4 Table. Results of IBDseq analysis of high-coverage data for 3 individuals per genotype per population.** Group 1 indicates the genotype and population for focal group 1, and Group 2 indicates the genotype and population for focal group 2. CHPL, Chapulhuacanito; STAC, Santa Cruz; COAC, Coacuilco; HUIC, Huichihuayán; PTHC, Puente de Huichihuayán. Comparison type indicates whether individuals were sampled from the same or different geographical locations. Number of tracts is the number of IBD segments identified by IBDseq. The median length of these tracts in basepairs and the upper 95% quantile is also reported for cases with a sufficient number of tracts.
(DOCX)

**S5 Table. Results of the analysis of the correlation between local minor parent ancestry in windows of varying size versus the average recombination rate in that window.** Windows of 100–500 kb are reported here. Since both ancestry and recombination rate are spatially correlated along the genome, for each analysis, we thinned the data to include 1 window per Mb, such that all analyses presented below include the same number of windows. Population names in italics indicate the focal samples primarily discussed in the main text.
(DOCX)

**S6 Table. Results of the analysis of the partial correlation between local minor parent ancestry and the number of linked coding and conserved basepairs.** Since recombination rate has a strong impact on local ancestry (S5 Table), this analysis attempts to control for this effect by accounting for local recombination rate. We repeated this analysis in windows of a particular genetic length (S7 Table). As before, we thinned windows to include only 1 per Mb. Population names in italics indicate the focal samples primarily discussed in the main text.
(DOCX)

**S7 Table. Analysis of correlations between minor parent ancestry and the number of coding or conserved basepairs in a window of a given genetic size.** Since recombination rate has a strong effect on minor parent ancestry, and recombination rate is strongly correlated with functional elements in swordtails (see Text F in S1 File), analyses in genetic windows allows us to disentangle these effects. Windows were thinned so that on average on window per Mb was retained, and the same number of windows was retained for each analysis. Population names in italics indicate the focal samples primarily discussed in the main text.
(DOCX)

**S8 Table. Summary of local ancestry correlations across populations in windows of different sizes.** Analysis was conducted on thinned data, so that for each data set 1 window per Mb was retained, resulting in the same number of windows across comparisons. Both *X. birchmanni × X. cortezi* (Santa Cruz, Chapulhuacanito, Huextetitla) and *X. birchmanni × X. malinche* populations (Acuapa, Aguazarca, Tlatemaco) are included in this analysis. Note that the *X. birchmanni × X. cortezi* populations at Santa Cruz and Huextetitla occur in the same river system. All *X. birchmanni × X. malinche* populations occur in different river systems. (DOCX)

**S9 Table. Summary of local ancestry correlations across populations in windows of different genetic size.** Binning into windows of varying genetic size allows us to partially control for the strong relationship between minor parent ancestry and recombination rate. We performed 2 analyses using a partial correlation approach, one comparing ancestry across populations while accounting for covariance due to the number of coding basepairs in a window, and a separate analysis comparing ancestry across populations while accounting for covariance due to the number of conserved basepairs in a window. Both *X. birchmanni × X. cortezi* (Santa Cruz, Chapulhuacanito, Huextetitla) and *X. birchmanni × X. malinche* populations (Acuapa, Aguazarca, Tlatemaco) are included in this analysis. Note that the *X. birchmanni × X. cortezi* populations at Santa Cruz and Huextetitla occur in the same river system. All *X. birchmanni × X. malinche* populations occur in different river systems. Analysis was conducted on thinned data, so that for each data set 1 window per 1.5 cMs was retained, resulting in the same number of windows analyzed across comparisons. (DOCX)

**S10 Table. Results of analyses controlling for potential artifacts.** Given the high correlations in minor parent ancestry observed between independently formed hybrid populations at Santa Cruz and Chapulhuacanito, we performed a series of analyses to exclude the role of potential artifacts in generating the signal we observe (see Methods). Thinned ancestry informative sites—ancestry informative sites were thinned before the HMM was run to better control for variation in power to infer ancestry across different regions of the genome. Intervals where ancestry informative sites occurred at higher than the median density were thinned to the median density and the HMM was re-run. Low power windows removed—windows with the fewest ancestry informative sites (lowest 5% quantile) were removed before analysis. Chromosome ends removed—windows within the last Mb of chromosomes, some of which overlapped with telomeric sequences, were removed. Repeats excluded—ancestry informative sites that overlapped with annotated repeats were excluded before analysis. All analyses were conducted on spatially thinned windows, so that for each data set 1 window per Mb was retained, resulting in the same number of windows across comparisons. (DOCX)

**S11 Table. Genes found in shared minor parent islands.** Bolded rows indicate islands that changed significantly in ancestry over time at $p < 0.05$. (XLSX)

**S12 Table. Identity of mitochondrial haplotypes found in both *X. birchmanni × X. cortezi*** populations and in time series data from Chapulhuacanito. (DOCX)

**S13 Table. Data from *Xiphophorus* species that was used to infer ancestral sequence states for LDhelmet.** (DOCX)

**S14 Table. Summary statistics input for ABCreg demographic inference analysis of Chapulhuacanito and Santa Cruz populations.**
(DOCX)

**S15 Table. Significantly enriched gene otology categories identified among genes in minor parent islands.**
(XLSX)

**S1 Fig. Posterior distributions for demographic parameters from ABCreg analysis for Santa Cruz and Chapulhuacanito populations.** We obtain well-resolved posterior distributions for most parameters, including migration rate from the *X. cortezi* parent (**A**), migration rate from the *X. birchmanni* parent (**B**), initial admixture proportion (**C**), and generations since initial hybridization (**D**). We did not recover well-resolved posterior distributions for hybrid population size where we essentially recover the prior distribution (a uniform distribution from 2–10,000; not plotted—see Dryad repository doi:10.5061/dryad.qnk98sfq1). The dotted lines indicated the maximum a posteriori estimate for that parameter. The data underlying this figure can be found in Dryad repository doi:10.5061/dryad.qnk98sfq1.
(PDF)

**S2 Fig. Results of principal component analysis of individuals sequenced to high coverage from Santa Cruz and Chapulhuacanito as well as from allopatric parental populations including all variable sites across the genome.** Analyses in the main text show PCAs based on particular ancestry tracts (i.e., homozygous *X. cortezi* or homozygous *X. birchmanni*; Fig 1). This analysis includes all variant sites from all ancestry tracts. *X. cortezi* and *X. birchmanni* separate along PC1. Hybrids cluster close to *X. cortezi* along PC1, as expected given that they derive >70% of their genome from *X. cortezi*. We include pure *X. birchmanni* individuals found sympatrically with hybrids from Santa Cruz and Chapulhuacanito (xbirCHPL and xbirSTAC), as well as parental individuals from allopatric populations (xbirCOAC, xcorHUIC, and xcorPTHC). Hybrids from the Santa Cruz and Chapulhuacanito populations separate from each other in this analysis (xbirXcorSTAC and xbirXcorCHPL codes in the plot, respectively). The data underlying this figure can be found in Dryad repository doi:10.5061/dryad.qnk98sfq1.
(PDF)

**S3 Fig. Additional analyses of population genetic patterns in high-coverage whole genome sequence data from Santa Cruz and Chapulhuacanito.** (**A**) We identified homozygous *X. birchmanni* ancestry tracts found in both Santa Cruz (STAC) and Chapulhuacanito (CHPL). We also subsampled these same regions from sympatric and allopatric pure parental individuals (*X. birchmanni* at COAC, CHPL, and STAC, *X. cortezi* at HUIC and PTHC). We found that Dxy to the reference *X. birchmanni* sequence was similar across the sequences analyzed for *X. birchmanni* tracts from hybrids and for those tracts from pure *X. birchmanni* individuals. As expected, *X. cortezi* populations had elevated Dxy to the *X. birchmanni* reference. (**B**) Similar patterns are observed for *X. cortezi* ancestry tracts analyzed with the same approach. *X. birchmanni*-derived regions had uniformly elevated Dxy compared to the *X. cortezi* reference (derived from the Puente de Huichihuayán or PTHC populations). Notably, homozygous *X. cortezi* ancestry tracts in hybrid individuals (*X. cor* × *X. bir* STAC and CHPL) have slightly but significantly higher Dxy to the *X. cortezi* reference sequence, hinting that the source population in both hybrid populations was somewhat diverged from the allopatric *X. cortezi* populations we sampled (consistent with expectations from geography). For both panels, colored points show the raw data and the black point and whiskers show the mean ± 2 standard errors of the mean. The data underlying this figure can be found in Dryad repository doi:10.5061/

**S4 Fig. Additional analyses of population genetic patterns in high-coverage whole genome sequence data from Santa Cruz and Chapulhuacanito focusing on patterns of nucleotide diversity.** In (**A**), we identified regions of the genome that were homozygous for *X. birchmanni* ancestry in the 3 deep-sequenced hybrids from Chapulhuacanito and the 3 from Santa Cruz. We subset these regions from the hybrids as well as allopatric parental populations and sympatric *X. birchmanni* individuals (*X. birchmanni* at COAC, CHPL, and STAC, *X. cortezi* at HUIC and PTHC). We find that pure *X. birchmanni* individuals have similar estimated π in these ancestry tracts, whereas Chapulhuacanito and to a lesser extent Santa Cruz have elevated π. In (**B**), we identified regions of the genome that were homozygous for *X. cortezi* ancestry in the 3 deep-sequenced hybrids from Chapulhuacanito and the 3 from Santa Cruz. We subset these regions from the hybrids as well as allopatric parental populations and sympatric *X. birchmanni* individuals. We found that although hybrids at Chapulhuacanito have similar estimated π to the *X. cortezi* parent in these ancestry tracts, hybrids at Santa Cruz have markedly lower estimated π. This may point to differences in the demographic history of Santa Cruz hybrids since the population formed, or in the *X. cortezi* parental population that contributed to the hybrids. For both panels, colored points show the raw data and the black point and whiskers shows the mean ± 2 standard errors of the mean. The data underlying this figure can be found in Dryad repository doi:10.5061/dryad.qnk98sfq1.
(PDF)

**S5 Fig. Comparisons of simulated versus observed mismatch statistics.** In homozygous *X. cortezi* ancestry tracts identified in high-coverage individuals, we observe high "mismatch" in homozygous alleles across the 2 hybrid populations relative to the values we observe within populations (Fig 1G; see Methods). We performed simulations to explore whether the level of mismatch observed across the 2 hybrid populations is compatible with a partially shared demographic history in the 2 populations (see Text C in S1 File for details). We modeled a population under demographic parameters inferred from previous simulations for Santa Cruz and Chapulhuacanito except that we modeled 10 (**A**), 50 (**B**), or 100 (**C**) generations of shared evolution. For each scenario, we repeated simulations for a total of 100 replicates (gray distributions) and compared mismatch values in these simulations to the mismatch value we calculated for the empirical data (see also Fig 1G). The simulated scenarios of partially shared evolutionary history overlapped with the empirical mismatch statistics for within-population analysis of Santa Cruz (green line) and Chapulhuacanito (blue line) but not the observed value for the between population mismatch statistics (yellow dashed line). This suggests that scenarios of non-independence in the 2 hybrid populations are not consistent with our data. The data underlying this figure can be found in Dryad repository doi:10.5061/dryad.qnk98sfq1.
(PDF)

**S6 Fig. Heatmap of the genetic relatedness matrices across different subsets of samples and regions of the genome.** Analysis was performed with GCTA (see Methods). (**A**) Analysis of *X. cortezi* ancestry tracts in 3 hybrids from the Santa Cruz (Hyb-STAC) population and 3 hybrids from Chapulhuacanito (Hyb-CHPL) population along with the same regions sampled from individuals from the 2 *X. cortezi* parental populations, Huichihuayán (HUIC) and Puente de Huichihuayán (PTHC). Cooler colors indicate lower genetic relatedness than the average in the data set and warmer colors indicate higher genetic relatedness than the average in the data set. Overall, we do not see evidence of genetic relatedness across hybrid populations but substantial evidence of relatedness within populations, with higher relatedness in comparisons

between Santa Cruz individuals. High relatedness between pure *X. cortezi* in Huichihuayán and Puente de Huichihuayán is not surprising since these populations occur on the same river. (**B**) Results of the same analysis, but for ancestry tracts that were homozygous for *X. birchmanni* in the 3 high-coverage hybrids from Santa Cruz (Hyb-STAC) and Chapulhuacanito (Hyb-CHPL), as well as the same regions sampled from pure *X. birchmanni* from the Río Santa Cruz (Xbir-STAC, Xbir-HUEX) and Chapulhuacanito (Xbir-CHPL) and from a pure *X. birchmanni* source population (Xbir-COAC). Overall, the results for hybrid populations are concordant with the analysis in **A**, with evidence of relatedness within but not between hybrid populations. However, we do observe more relatedness across *X. birchmanni* individuals from different rivers than we expected a priori. See Text E in S1 File for a discussion of these results. The data underlying this figure can be found in Dryad repository doi:10.5061/dryad.qnk98sfq1.
(PDF)

**S7 Fig. Frequency and length of IBD tracts inferred via IBDseq analysis for different comparisons of data from high coverage individuals.** (A) The vast majority of IBD tracts identified by our analysis corresponded to IBD tracts shared between individuals sampled from the same geographical location. These also represented the longest IBD tracts on average. (B) Many fewer IBD tracts were identified between individuals originating from different geographical locations, and these tracts were shorter on average. The majority of these "between-population" tracts were attributable to tracts shared with pure *X. birchmanni* or pure *X. cortezi* populations (see Text D in S1 File, S4 Table). (C) Very few IBD tracts were identified between the 2 hybrid populations, and these were on average approximately 20 kb in length. This suggests that there is little detectable genetic exchange or shared demographic history between these populations. Moreover, since this length is smaller than the typical ancestry tract in either population, these tracts likely are identical-by-descent due to events that preceded hybridization between *X. birchmanni* and *X. cortezi* in either hybrid population. Red dashed line shows median of each distribution. The data underlying this figure can be found in Dryad repository doi:10.5061/dryad.qnk98sfq1.
(PDF)

**S8 Fig. Frequency and length of IBD tracts inferred via IBDseq analysis within hybrid populations.** Plots show the (**A**) Chapulhuacanito (CHPL) or (**B**) Santa Cruz (STAC) hybrid populations. Red dashed line shows median of each distribution. For other comparisons within and between populations, see S4 Table. The data underlying this figure can be found in Dryad repository doi:10.5061/dryad.qnk98sfq1.
(PDF)

**S9 Fig. Comparisons of inferred local recombination rates in *X. birchmanni* and *X. cortezi*.** Since *Xiphophorus* species lack a PRDM9 ortholog that is active in meiosis and recombine primarily at functional elements that are largely conserved between species (S10 Fig), we expected their recombination maps to be conserved as well. Comparison of local recombination rates across a range of window sizes (**A–C**) confirms this prediction (**A**–Spearman's $\rho = 0.55$; **B**–Spearman's $\rho = 0.57$; **C**–Spearman's $\rho = 0.62$). Given that previous simulations of the *X. birchmanni* recombination map indicated that the expected correlations between the true recombination map and the inferred LD-based recombination map were approximately 0.65 in 50 kb windows [1], this suggests that the 2 maps may be nearly identical. See Text F in S1 File for more detail. The data underlying this figure can be found in Dryad repository doi:10.5061/dryad.qnk98sfq1.
(PDF)

**S10 Fig. Recombination rate as a function of distance to certain genomic elements.** Average recombination rate ($\rho$/bp) in sliding 5 kb windows as a function of distance of that window from the nearest transcriptional start site (TSS) or H3K4me3 peak in testis tissue in basepairs in *X. birchmanni* (**A**, **C**) and *X. cortezi* (**B**, **D**). In both species, average recombination rates peak near the TSS (**A**, **B**), as frequently observed in species that lack a PRDM9 ortholog active in specifying recombination hotspots. This pattern is thought to be driven by recombination machinery defaulting to the locations of existing H3K4me3 marks in the absence of PRDM9-driven H3K4me3 marks. Indeed, we see that both species have elevated recombination near shared H3K4me3 identified in *X. birchmanni* testis samples (data reanalyzed from [1]; **C**, **D**). Gray lines show individual replicates bootstrap resampling the data, with a total of 500 replicates plotted. Blue line shows the average across simulations. The data underlying this figure can be found in Dryad repository doi:10.5061/dryad.qnk98sfq1.
(PDF)

**S11 Fig. Average decay of admixture linkage disequilibrium among hybrid individuals from the Chapulhuacanito (left) and Santa Cruz (right) hybrid populations.** The y-axis shows the average $R^2$ value in sliding 5 kb windows and the x-axis shows the physical distance between ancestry informative sites in the focal window. Both populations asymptote to background levels of admixture LD by approximately 500 kb, although we thin our data by 1 Mb to be conservative (see Methods). We note also that there is an excess of admixture linkage disequilibrium in both populations where the minimum average $R^2$ greatly exceeds one over the number of sampled individuals, particularly in the Santa Cruz population. This is suggestive of population structure, a recent pulse of migration, or assortative mating in these populations (see [68] for data on assortative mating in Santa Cruz). The data underlying this figure can be found in Dryad repository doi:10.5061/dryad.qnk98sfq1.
(PDF)

**S12 Fig. Visualization of cross-population versus within population (or river) correlations in minor parent ancestry.** Average minor parent ancestry for matched regions of the genome is plotted on the x and y axes in 100 kb windows. (**A**) Average ancestry in Chapulhuacanito (CHPL) population samples from 2021 are plotted against average ancestry in the same windows from the Santa Cruz (STAC) population sampled in 2020. Correlations in local ancestry are high, but windows with high minor parent ancestry in Chapulhuacanito and low minor parent ancestry in Santa Cruz (and vice versa) can be visualized in the plot. By contrast, local ancestry comparisons from the same population but different years (**B**) or 2 nearby populations in the same river drainage (**C**) are remarkably concordant. **D** and **E** show results from subsampling 20 individuals from the same population and sampling year. Again, correlations between individuals sampled from the same population (**D**–Chapulhuacanito, **E**–Santa Cruz) greatly exceed observed correlations between drainages (**A**). The data underlying this figure can be found in Dryad repository doi:10.5061/dryad.qnk98sfq1.
(PDF)

**S13 Fig. Performance of local ancestry inference in parental individuals.** (**A**) Analyses of genome-wide ancestry in allopatric *X. birchmanni* individuals demonstrate that *ancestryinfer* correctly infers that these individuals are homozygous for *X. birchmanni* ancestry across the genome. (**B**) This result can also be observed by examining local ancestry in the same individuals (3 representative individuals plotted here), where entire chromosomes are inferred to have 2 *X. birchmanni*-derived and no *X. cortezi*-derived haplotypes. (**C**) Similarly, ancestry inference in allopatric *X. cortezi* individuals is accurate, with all individuals inferred to be homozygous *X. cortezi* throughout the genome. (**D**) This result can also be observed by examining

local ancestry in the same individuals (3 representative individuals plotted here), where entire chromosomes are inferred to have 2 *X. cortezi* derived and no *X. birchmanni*-derived haplotypes. The data underlying this figure can be found in Dryad repository doi:10.5061/dryad.qnk98sfq1.
(PDF)

**S14 Fig. Analyses of genome-wide ancestry in hybrid offspring of known crosses.** (**A**) $F_1$ hybrids produced in lab are inferred by *ancestryinfer* to have precisely 50% of their genome derived from *X. birchmanni* and 50% from *X. cortezi*, with essentially no variation in ancestry observed, as expected for this cross. (**B**) Three representative individuals showing local ancestry across chromosome 1 in $F_1$ hybrids. $F_1$ hybrids are inferred to have 1 *X. birchmanni* and 1 *X. cortezi* haplotype across the chromosome, as expected from the cross design. (**C**) $F_2$ hybrids produced in lab are inferred to have on average 50% of their genome derived from each parental species, but with substantial variation in genome-wide ancestry induced by recombination between the *X. birchmanni* and *X. cortezi* haplotypes in their $F_1$ parents, followed by independent assortment. (**D**) Local ancestry across chromosome 1 for 3 representative $F_2$ individuals. (**E**) Local ancestry across chromosome 1 for 4 representative $BC_1$ individuals ($F_1$ hybrids crossed to *X. cortezi*). We produced fewer backcross individuals than other cross types so do not plot genome-wide ancestry for these individuals. The data underlying this figure can be found in Dryad repository doi:10.5061/dryad.qnk98sfq1.
(PDF)

**S15 Fig. Estimates of error rate using *mixnmatch* simulations.** As a complementary analysis to ancestry inference in known crosses, we used the program *mixnmatch* to simulate late generation hybrids with parameters matching the Chapulhuacanito (**A**) and Santa Cruz (**B**) populations to estimate our error rate in the case of late generation hybrids between *X. cortezi* and *X. birchmanni*. (**A**) Simulations of Chapulhuacanito hybrids using the MAP estimates of demographic parameters from the posterior distributions generated by ABCreg and with a simulated hybrid population size of 5,000. Accuracy in local ancestry inference for 50 simulated hybrid individuals is shown here. The average error rate in these simulations was 0.3%. (**B**) Simulations of Santa Cruz using the MAP estimates of demographic parameters from the posterior distributions generated by ABCreg and with a simulated hybrid population size of 5,000. Accuracy in local ancestry inference for 50 simulated hybrid individuals is shown here. The average error rate in these simulations was 0.4%. As expected, the error rate is slightly higher for simulations of Santa Cruz, since this population is likely to be older and smaller ancestry tracts are more difficult to accurately infer. The data underlying this figure can be found in Dryad repository doi:10.5061/dryad.qnk98sfq1.
(PDF)

**S16 Fig. Additional results of wavelet decomposition analyses focusing on the relationship between minor parent ancestry and recombination rate.** (**A**) Spatial wavelet decomposition of the overall Pearson correlation between inferred minor parent ancestry in Chapulhuacanito (CHPL) and Santa Cruz (STAC) versus the inferred recombination rate, measured at a resolution of 1 kb. (**B**) The contribution of a given spatial scale to the overall correlation is a weighted correlation of wavelet coefficients for the 2 signals at that scale, weighted by the variances in each signal at that scale, also obtained from the discrete wavelet transform. We show this decomposition for Chapulhuacanito and note that the general pattern is highly similar for Santa Cruz (STAC). The contributions sum to the total Pearson correlation calculated at the finest resolution of measurement (1 kb). For simplicity, we omit "scaling" variances which are leftover variance due irregularity of chromosome lengths. The data underlying this figure can

be found in Dryad repository doi:10.5061/dryad.qnk98sfq1.
(PDF)

**S17 Fig. Additional results of wavelet decomposition analyses focusing on the relationship between minor parent ancestry in different hybrid populations.** (**A**) Wavelet correlations between minor parent ancestry proportion in cross population comparisons between hybrids derived from the same hybridizing pair of *X. birchmanni* × *X. cortezi* (CHPL vs. STAC), the same hybridizing pair of *X. birchmanni* × *X. malinche* populations (ACUA vs. AGZC), and between hybrids from different hybridizing pairs (CHPL vs. ACUA). Points are weighted averages across chromosomes with error bars representing 95% jackknife confidence intervals. For visualization, we omit the confidence interval for the wavelet correlation of ancestry in ACUA vs. AGZC at the largest scale, since it is large and overlaps zero. See discussion in Text J in S1 File regarding fine scale correlations (e.g., 1 kb). (**B**) Here, the correlations in **A** are weighted by the variances in ancestry in each population at a given scale to reflect the contribution of that scale to the overall correlation. (**C**) Partial wavelet correlations of minor parent ancestry between populations after accounting for variation in recombination. At each scale, we compute a wavelet correlation of the residuals for each population from a linear model with recombination as a predictor. Using these values, we find a comparatively larger reduction in comparisons across hybrid population types (i.e., CHPL vs. ACUA). This would be consistent with a larger portion of the ancestry correlation in these comparisons being driven by the shared effects of recombination. The data underlying this figure can be found in Dryad repository doi:10.5061/dryad.qnk98sfq1.
(PDF)

**S18 Fig. Observed shared minor parent deserts and islands across hybrid population types.** Minor parent deserts and islands were identified in the *X. birchmanni* × *X. cortezi* hybrid populations Chapulhuacanito (CHPL) and Santa Cruz (STAC). We then asked whether these regions overlapped with minor parent deserts or islands found in *X. birchmanni* × *X. malinche* hybrid populations (Acuapa–ACUA, Aguazarca–AGZC, and Tlatemaco–TLMC). Black diamonds indicate the observed number of overlaps between minor parent deserts and islands in any *X. birchmanni* × *X. malinche* population and the *X. birchmanni* × *X. cortezi* hybrid populations (with the number shared between Chapulhuacanito and Santa Cruz shown in blue for comparison). Colored points show the number of shared deserts or islands from replicates jack-knife bootstrapping the genome in 10 cM windows. Gray points show null expectations for each comparison (see Methods). As expected, we observe many more shared deserts and islands among *X. birchmanni* × *X. cortezi* hybrid populations than between *X. birchmanni* × *X. cortezi* and *X. birchmanni* × *X. malinche* hybrid populations. The data underlying this figure can be found in Dryad repository doi:10.5061/dryad.qnk98sfq1.
(PDF)

**S19 Fig. Circos plot of the new *X. cortezi* genome assembly (green) relative to *X. birchmanni* (red).** Inversions between the 2 species are highlighted by light red lines connecting chromosomes, while co-linear regions are connected by light green lines (based on minimap2 alignments). Gray boxes show schematics of each chromosome. Black circles on the chromosome schematic indicate locations on the chromosomes where telomeric sequences were detected (using seqtk telo). The Circos plot was generated by the R package circlize. Note that several putative translocations are identified, which we treat with caution in the absence of Hi-C data for *X. cortezi*. The data underlying this figure can be found in Dryad repository doi:10.5061/dryad.qnk98sfq1.
(PDF)

**S20 Fig. Plot of average minor parent ancestry across the first 6 Mb of chromosome 13 in both Santa Cruz and Chapulhuacanito with the location of *ndufs5* highlighted.** *ndufs5* (red line) is involved in a known mitonuclear incompatibility (see main text) but the region surrounding it was not identified as a shared desert (found as a desert in Chapulhuacanito but not Santa Cruz). However, examination of local ancestry indicates that *X. birchmanni* ancestry at *ndufs5* (red line) is rare in both populations, but was slightly higher than the lowest 5% quantile for Santa Cruz (lowest 5% quantile for *X. birchmanni* ancestry is 2.2% in Santa Cruz, versus 2.3% *X. birchmanni* ancestry observed at *ndufs5*). The data underlying this figure can be found in Dryad repository doi:10.5061/dryad.qnk98sfq1.
(PDF)

**S21 Fig. Comparisons of minor parent ancestry across genes of different annotations in the 2 hybrid populations.** In the "Complexes vs. all proteins" comparison, we calculated average minor parent ancestry in genes involved in protein complexes (see Text M in S1 File) in both Chapulhuacanito and Santa Cruz (colored points and whiskers), compared to null data sets generated from randomly sampling matched sets of protein coding genes (gray points). Observed minor parent ancestry in these proteins is slightly but significantly lower than the null distribution. We next considered the possibility that this signal was driven by genes involved in protein complexes being highly conserved. We identified all genes that had a single reciprocal best blast hit with human proteins ("1:1 orthologs"), excluded genes involved in protein complexes, and repeated the analysis (see Text M in S1 File). We found that these genes were similarly depleted in minor parent ancestry relative to the null, suggesting that the signal we observe with genes in protein complexes is driven by conservation, not directly by their role in protein–protein interactions. The data underlying this figure can be found in Dryad repository doi:10.5061/dryad.qnk98sfq1.
(PDF)

**S22 Fig. Average *X. birchmanni* ancestry in Santa Cruz and Chapulhuacanito surrounding 2 new regions identified as under strong selection in F$_2$ hybrids between *X. birchmanni* and *X. cortezi*.** Results in **A** show identified region on chromosome 7 and results in **B** show identified region on chromosome 14. Each point indicates average minor parent ancestry in a 0.05 cM window in Santa Cruz or Chapulhuacanito. Red line indicates the region identified as under selection in F$_2$ hybrids between *X. birchmanni* and *X. cortezi* and black line in **B** indicates the location of a shared minor parent ancestry desert between Santa Cruz and Chapulhuacanito on chromosome 14. While the region on chromosome 7 does not overlap with a shared minor parent ancestry desert, it does overlap with an ancestry desert in Santa Cruz at approximately 16.5 Mb, and *X. birchmanni* ancestry across the larger region from 16–22 Mb is relatively low in both Santa Cruz and Chapulhuacanito (in the 15% and 10% quantile of minor parent ancestry genome wide, respectively). These regions are exciting candidates for identifying additional hybrid incompatibilities between *X. birchmanni* and *X. cortezi*. The data underlying this figure can be found in Dryad repository doi:10.5061/dryad.qnk98sfq1.
(PDF)

**S23 Fig. Results of admix'em simulations with selection.** Plots show genome-wide admixture proportions at the end of simulations described in Text G in S1 File. **A** and **C** show simulations using demographic parameters inferred for the Santa Cruz population from ABCreg, with selection on 40 recessive hybrid incompatibilities (**A**) or on 40 hybrid incompatibilities with a range of dominance coefficients (**C**). The dotted gray line shows the observed admixture proportion in Santa Cruz. **B** and **D** show simulations using demographic parameters inferred for the Chapulhuacanito hybrid population from ABCreg, with selection on 40 recessive

hybrid incompatibilities (**B**) or on 40 hybrid incompatibilities with a range of dominance coefficients (**D**). The dotted gray line shows the observed admixture proportion in Chapulhuacanito. For all sets of admix'em simulations, we had to modify the initial admixture proportion from the posterior distributions inferred by ABCreg because simulation selection dramatically shifts the initial admixture proportion (see Text G in S1 File). As a result, we wanted to confirm that after simulations of selection with admix'em, the final admixture proportions roughly matched those observed in the Santa Cruz and Chapulhuacanito hybrid populations. The data underlying this figure can be found in Dryad repository doi:10.5061/dryad.qnk98sfq1. (PDF)

**S24 Fig. Results of admix'em simulations modeling shared selection in 2 independently formed hybrid populations.** For each simulation, we drew demographic parameters from the posterior distributions inferred by ABCreg for Santa Cruz and Chapulhuacanito. We simulated 2 hybrid populations using these demographic parameters as described in Text G in S1 File. In one set of simulations, we implemented selection on 40 pairs of recessive hybrid incompatibilities, with selection coefficients drawn from an exponential distribution with a mean of 0.6. The observed cross-population correlations in ancestry in 250 kb windows for 50 replicates of this simulation scenario are shown in (**A**). In another set of simulations, we implemented selection on 40 pairs of hybrid incompatibilities as before, except that we drew dominance coefficients from a uniform distribution of 0–0.5 and selection coefficients from an exponential distribution with a mean of 0.4. The observed cross-population correlations in ancestry in 250 kb windows for 50 replicates of this simulation scenario are shown in (**B**). The inferred cross-population correlation coefficient in 250 kb windows between Santa Cruz and Chapulhuacanito in shown by the dotted gray line. Although the observed value falls in the upper range of the distribution inferred from simulations, these results indicate that selection, combined with the inferred demographic history of these 2 populations, can drive the high correlations in local ancestry we observe. (**C**) Neutral simulations with no selection implemented are plotted for comparison (see Text G in S1 File). The data underlying this figure can be found in Dryad repository doi:10.5061/dryad.qnk98sfq1. (PDF)

**S25 Fig. Analysis of expected performance of ABCreg.** To evaluate the expected performance of ABCreg, we randomly sampled 100 simulations generated for ABC demographic inference. We treated the summary statistics from each simulation as if it were the real data and ran ABCreg. We asked for a given simulation, whether the true value for the focal parameter fell within the 50% quantile of the posterior distribution generated by ABCreg (top) or the 95% quantile (bottom). Plotted are the proportion of simulations where the true value for each parameter fell within the 50% or 95% quantile of the posterior distribution generated by ABCreg. For all parameters, the true value is very likely to fall in the 95% quantile of the posterior distribution generated by ABCreg. The data underlying this figure can be found in Dryad repository doi:10.5061/dryad.qnk98sfq1. (PDF)

**S26 Fig. Alternative analysis of enrichment of shared deserts and islands across hybrid populations.** The number of shared deserts (left) and islands (right) between the Santa Cruz (STAC) and Chapulhuacanito (CHPL) populations compared to expectations by chance, using a different permutation approach that preserves the structure of local ancestry variation along the genome (see Methods). Gray points and boxplots indicate expectations from permutations; black points indicate the observed data. The data underlying this figure can be found in Dryad

**S27 Fig. Inferred separation between source populations of the 2 hybrid populations at Santa Cruz and Chapulhuacanito based on ABC simulations implemented in SLiM.** Simulations of shared hybrid population history are not consistent with population genetic statistics calculated from our data (Text C in S1 File). We used additional simulations to explore what source population demographic history might be consistent with our data in and ABC framework. Shown here are the results of those simulations; see Text C in S1 File for simulation details. (**A**) Posterior distribution of generations of genetic drift between the source *X. cortezi* populations contributing to Santa Cruz and Chapulhuacanito based on 500 accepted simulations. Red line indicates the MAP estimate of 1,315 generations (95% credible intervals: 329–3,099). (**B**) Posterior distribution of post-split population size 1 based on 500 accepted simulations. Red line indicates the MAP estimate of 2,187 individuals (95% credible intervals: 233–4,851). (**C**) Posterior distribution of post-split population size 2 based on 500 accepted simulations. Red line indicates the MAP estimate of 979 individuals (95% credible intervals: 193–4,816). The data underlying this figure can be found in Dryad repository doi:10.5061/dryad.qnk98sfq1.
(PDF)

**S28 Fig. Example of a likely error in local ancestry inference detected in an $F_1$ hybrid between *X. birchmanni* and *X. cortezi* on chromosome 3.** In this individual, the local ancestry call switches from heterozygous for ancestry to homozygous *X. birchmanni* for 11 kb. While errors like these are occasionally detected in our data sets of parental and $F_1$ hybrid individuals, we estimate our overall error rates for these individuals to be approximately 0.1% per ancestry informative site. The data underlying this figure can be found in Dryad repository doi:10.5061/dryad.qnk98sfq1.
(PDF)

**S29 Fig. Simulations of power to detect selected sites in populations matching the demographic history of Santa Cruz (STAC) and Chapulhuacanito (CHPL) populations.** For each population and selection coefficient (s = 0.01–0.1), we performed 100 pairs of simulations of selection using admix'em. For each pair of simulations, we randomly identified the location of the site under selection and then performed simulations with demographic parameters drawn from the ABCreg posterior distributions for Santa Cruz and Chapulhuacanito, respectively. We then identified minor parent deserts as we had for the real data (see Methods) and determined the proportion of time over 100 simulations that we correctly identified the ancestry desert in each population. The data underlying this figure can be found in Dryad repository doi:10.5061/dryad.qnk98sfq1.
(PDF)

## Acknowledgments

We thank Yaniv Brandvain, Kelley Harris, Priya Moorjani, and members of the Schumer lab and Coop lab for helpful comments on earlier versions of this manuscript. We are grateful to the Federal Government of Mexico for permission to collect fish. Stanford University and the Stanford Research Computing Center provided computational support for this project.

## Author Contributions

**Conceptualization:** Quinn K. Langdon, Carla Gutiérrez-Rodríguez, Molly Morris, Molly Schumer.

**Data curation:** Quinn K. Langdon, Stepfanie M. Aguillon, Daniel L. Powell, Theresa Gunn, John J. Baczenas, Alex Donny, Kang Du, Oscar Ríos-Cárdenas, Carla Gutiérrez-Rodríguez, Molly Morris.

**Formal analysis:** Quinn K. Langdon, Jeffrey S. Groh, Stepfanie M. Aguillon, Daniel L. Powell, Cheyenne Payne, Tristram O. Dodge, Kang Du, Molly Schumer.

**Funding acquisition:** Manfred Schartl, Molly Schumer.

**Investigation:** Jeffrey S. Groh, Stepfanie M. Aguillon, John J. Baczenas, Alex Donny, Tristram O. Dodge, Oscar Ríos-Cárdenas, Carla Gutiérrez-Rodríguez, Molly Morris.

**Methodology:** Quinn K. Langdon, Daniel L. Powell, Theresa Gunn, John J. Baczenas, Alex Donny, Kang Du, Oscar Ríos-Cárdenas, Carla Gutiérrez-Rodríguez, Molly Morris, Molly Schumer.

**Project administration:** Theresa Gunn, Manfred Schartl, Carla Gutiérrez-Rodríguez, Molly Morris, Molly Schumer.

**Resources:** Molly Schumer.

**Software:** Jeffrey S. Groh, Cheyenne Payne, Tristram O. Dodge, Kang Du.

**Validation:** Quinn K. Langdon, Manfred Schartl.

**Visualization:** Quinn K. Langdon, Molly Schumer.

**Writing – original draft:** Tristram O. Dodge, Molly Schumer.

**Writing – review & editing:** Jeffrey S. Groh, Tristram O. Dodge, Manfred Schartl, Oscar Ríos-Cárdenas, Carla Gutiérrez-Rodríguez, Molly Morris, Molly Schumer.

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
