## [Editor Report · Decision Letter 0]

17 Jan 2024

Dear Dr Schumer, 

Thank you for submitting your manuscript entitled "Genome evolution is surprisingly predictable after initial hybridization" for consideration as a Research Article by PLOS Biology.

Your manuscript has now been evaluated by the PLOS Biology editorial staff, as well as by an academic editor with relevant expertise, and I'm writing to let you know that we would like to send your submission out for external peer review.

Once your full submission is complete, your paper will undergo a series of checks in preparation for peer review. After your manuscript has passed the checks it will be sent out for review. To provide the metadata for your submission, please Login to Editorial Manager (https://www.editorialmanager.com/pbiology) within two working days, i.e. by Jan 19 2024 11:59PM.

Kind regards,

Roli Roberts

Roland Roberts, PhD

Senior Editor

PLOS Biology

rroberts@plos.org

---

## [Decision Letter · Decision Letter 1]

25 Mar 2024

Dear Molly,

Thank you for your patience while your manuscript "Genome evolution is surprisingly predictable after initial hybridization" was peer-reviewed at PLOS Biology. It has now been evaluated by the PLOS Biology editors, an Academic Editor with relevant expertise, and by two independent reviewers. We had recruited a third, but they failed to deliver their comments in a timely fashion; please accept our apologies for the resulting delay.

Reviewer #1 is largely positive, but is concerned that your evidence for the independence of the hybridisation events is weak, saying that “rigorously ruling out alternative demographic scenarios is needed.” S/he also asks about the verisimilitude of the SLiM simulations, and requests some statistical tests. Reviewer #2 thinks that the study is exceptionally well done, but questions whether the advance (especially over your own previous work) is sufficient for PLOS Biology.

We also asked the reviewers to cross-comment on each other's assessments. I discussed the reviews, and these cross-comments, with the Academic Editor, who had raised similar questions regarding the magnitude of advance to those raised by reviewer #2. On balance, we would like to consider your study further, but as a Short Report.

In light of the reviews, which you will find at the end of this email, we would like to invite you to revise the work to thoroughly address the reviewers' reports. IMPORTANT: As well as attending to the reviewers' concerns, please re-format your paper to a Short Report; in practice, this will simply involve reducing the number of Figures from 5 to 4, either by combining two of them, or by moving some material to the supplement. I have taken the liberty of already changing the article type to "Short Reports" in our system.

Given the extent of revision needed, we cannot make a decision about publication until we have seen the revised manuscript and your response to the reviewers' comments. Your revised manuscript is likely to be sent for further evaluation by all or a subset of the reviewers.

**IMPORTANT - SUBMITTING YOUR REVISION**

*Re-submission Checklist*

*Published Peer Review*

*PLOS Data Policy*

*Blot and Gel Data Policy*

Sincerely,

Roli

Roland Roberts, PhD

Senior Editor

PLOS Biology

rroberts@plos.org

REVIEWERS' COMMENTS:

Reviewer #1:

This manuscript utilizes independent Xiphophorus hybrid populations to understand if evolution after hybridization is predictable at the genomic level. The interest of the study is motivated by the growing understanding that hybridization and introgression have contributed to the evolution of many taxa across the tree of life. Yet we do not understand fully the consequences of hybridization and gene flow, because traditionally studies have focused on the consequences of hybridization for speciation and divergence. Now the focus is shifting, and the field is considering hybrids themselves as interesting evolutionary units. 

This is not the first study to investigate predictability of hybrid genome evolution, and few similar studies have been conducted in other taxa and also in Xiphophorus by the authors themselves. In fact, the current study builds on a wealth of investigations on the Xiphophorus hybrid system. It is this foundation and background knowledge that makes the findings of this study interesting, adding to our understanding of hybrid genome evolution and how hybridization can contribute to evolution in general. My main comment involves clearly demonstrating that the hybrid populations are of independent origin, which forms the basis of the core argument: independent populations evolve in repeated manner.

Main comments:

1. Did the authors rule out gene flow from parental species into the hybrid populations as one explanation of the observed patterns? For example, say the hybrid populations did not form independently, but from the same event. After separation each of them (or one) received gene flow from one of the parental species. This would create independent recombination breakpoints in the two hybrid populations even when they have the same origin. Then one might end up with repeatable ancestry tracts across hybrid populations but different recombination breakpoints (or ancestry transitions as the authors call them). Did the authors explicitly rule this out? Did they use demographic modelling to test which of the alternative scenarios best fits the data? How have the drainage systems changes in the past >100 generations? Even if they are separated now, is that stable since the formation of hybrids? How far can the fish disperse? Rigorously ruling out alternative demographic scenarios is needed. 

2. The SLiM simulations are the basis of the observation "more repeatable than expected by chance". Did the SLiM simulations incorporate the biological aspects of the model system (pop size, generation time, possible overlapping generations etc.)? If not, is it surprising that the observed parameters do not correspond the simulated ones? Complementing the analyses with simulations that account the demographic history and biological features of the system would make it clear the observed patterns are not expected under neutrality.

3. There are several passages in the main text where the authors state something is significantly different from random and refer to Figure and methods. It would be more convincing to refer to test statistic or give a short justification why something is different from random, rather than just stating it. e.g. lines 444-446, 453-455.

4. One of the interesting ideas in this study is that hybrid genome evolution is more repeatable the more distantly related the hybridizing species are. This is what one would expect if the number or strength of selection targets, e.g. incompatibilities, increases with increasing divergence. Results from the study are consistent with this idea. To be able to make a more general statement, one would need more data points on the divergence continuum, since the current study has only two data points (hybrids between closely related Xiphophorus species and hybrids between more distantly related Xiphophorus species). In this sense the results are tentative, but very interesting.

5. The authors have also used various techniques (e.g. Li et al. 2022) to infer the location of candidate DMIs. Does the location of these correlate with minor parent ancestry deserts?

This is a very interesting study with wealth of background information on the system, that significantly adds to our understanding of evolution after hybridization. Although some analyses require further elaboration, the study is mainly rigorously executed. The analysis pipelines are diverse and state-of-the-art, partly developed by the authors themselves. 

Reviewer #2:

This manuscript compares two independent populations of hybrids between Xiphophorus birchmanni and X. cortezi, and shows that patterns of local ancestry are strongly correlated between the two populations. The authors then apply a recently developed analytical technique to infer that the correlation results from strong selection in the first few generations after hybridization, and report follow-up analyses to identify the possible targets of selection.

The manuscript is exceptionally thorough and careful in its methodology, and includes well-designed tests to support key premises, such as the independent origin of the two hybrid populations and their respective ages. It's one of the first applications of wavelet analysis of ancestry blocks to study the history of genomic sorting after a pulse of hybridization. The analysis of minor parent ancestry "islands" and "deserts" appears to be highly effective at identifying genomic regions that have experienced selection in hybrids, based on simulations, temporal comparisons, and detection of a previously identified mitonuclear incompatibility between the parent species.

The overall conclusion of the paper extends previous research by the same group, particularly a study (Langdon et al 2022 PLOS Genetics) comparing two birchmanni x cortezi hybrid populations in the same river drainage to multiple birchmanni x malinche hybrid populations (birchmanii is more closely related to malinche than to cortezi). The 2022 study, and previous work by the same group focused on the birchmanni x malinche hybrid populations, found some degree of repeatability in patterns of local ancestry across multiple hybrid populations, and correlations between minor parent ancestry and genomic features such as recombination rate and gene density. The current study finds a higher level of repeatability, which is attributed to strong intrinsic incompatibilities between the genomes of X. birchmanni and X. cortezi. This study is likely to be of great interest to speciation genomics researchers, perhaps more for the comprehensive and innovative set of methods the authors employ than for the major conclusions. However, it may not have as much relevance to scientists outside this subfield, given the similarity to previous research on repeatability of ancestry patterns in replicated swordtail hybrid populations.

Minor specific comments:

L83-84: Punctuation error? The phrase "([15]; and the sites under selection are shared)" is hard to parse whether the condition applies to ref 15, the text preceding the citation, or both.

L146: Change ref [32] to [33]

L181: Change "manuscript refer" to "manuscript we refer"

---

## [Decision Letter · Decision Letter 2]

12 Jun 2024

Dear Molly,

Thank you for your patience while we considered your revised manuscript "Genome evolution is surprisingly predictable after initial hybridization" for publication as a Short Report at PLOS Biology. This revised version of your manuscript has been evaluated by the PLOS Biology editors, the Academic Editor and one of the original reviewers.

Based on the review and our Academic Editor's assessment of your revision, we are likely to accept this manuscript for publication, provided you satisfactorily address the following data and other policy-related requests:

IMPORTANT - please attend to the following:

a) Please change the title to "Swordtail fish hybrids reveal that genome evolution is surprisingly predictable after initial hybridization" (we think that this clarifies the study system and nature of the evidence, while retaining the generalisable findings).

b) I think that some of the panels in Fig 1 are mislabelled. And two supp Figs (S19, S26) look like they’re not displaying correctly. Please check and correct if necessary.

c) Please address my Data Policy requests below; specifically, we need you to supply the numerical values underlying Figs 1CDEF, 2ABCD, 3ABCD, 4ABCDE, S1ABCD, S2, S3AB, S4AB, S5ABC, S6AB, S7ABC, S8AB, S9ABC, S10ABCD, S11AB, S12ABCDE, S13ABCD, S14ABCDE, S15AB, S16AB, S17ABC, S18, S19, S20, S21, S22AB, S23ABCD, S24ABC, S25, S26, S27ABC, S28, S29, either as a supplementary data file or as a permanent DOI’d deposition.

d) Please cite the location of the data clearly in all relevant main and supplementary Figure legends, e.g. “The data underlying this Figure can be found in S1 Data” or “The data underlying this Figure can be found in https://zenodo.org/records/XXXXXXXX

e) I note that you mention the reviewers ("and two anonymous reviewers") in the Acknowledgements. While we appreciate the sentiment, this is against PLOS policy, so please could you remove this?

f) Please make any custom code available, either as a supplementary file or as part of your data deposition.

We expect to receive your revised manuscript within two weeks. 

*Published Peer Review History*

*Press*

Sincerely,

Roli

Roland Roberts, PhD

Senior Editor

rroberts@plos.org

PLOS Biology

DATA POLICY:

Regardless of the method selected, please ensure that you provide the individual numerical values that underlie the summary data displayed in the following figure panels as they are essential for readers to assess your analysis and to reproduce it: Figs 1CDEF, 2ABCD, 3ABCD, 4ABCDE, S1ABCD, S2, S3AB, S4AB, S5ABC, S6AB, S7ABC, S8AB, S9ABC, S10ABCD, S11AB, S12ABCDE, S13ABCD, S14ABCDE, S15AB, S16AB, S17ABC, S18, S19, S20, S21, S22AB, S23ABCD, S24ABC, S25, S26, S27ABC, S28, S29. NOTE: the numerical data provided should include all replicates AND the way in which the plotted mean and errors were derived (it should not present only the mean/average values).

CODE POLICY

Per journal policy, if you have generated any custom code during the curse of this investigation, please make it available without restrictions upon publication. Please ensure that the code is sufficiently well documented and reusable, and that your Data Statement in the Editorial Manager submission system accurately describes where your code can be found.

DATA NOT SHOWN?

REVIEWERS' COMMENTS:

Reviewer #1:

The authors have addressed my criticism with several new analyses that provide support to their results and conclusions. I congratulate the authors for a thorough investigation and insightful paper!

---

## [Editor Report · Decision Letter 3]

9 Jul 2024

Dear Dr Molly,

Thank you for the submission of your revised Short Report "Swordtail fish hybrids reveal that genome evolution is surprisingly predictable after initial hybridization" for publication in PLOS Biology. On behalf of my colleagues and the Academic Editor, Leonie Moyle, I'm pleased to say that we can in principle accept your manuscript for publication, provided you address any remaining formatting and reporting issues. These will be detailed in an email you should receive within 2-3 business days from our colleagues in the journal operations team; no action is required from you until then. Please note that we will not be able to formally accept your manuscript and schedule it for publication until you have completed any requested changes.

Sincerely,

Roli

Senior Editor

PLOS Biology

rroberts@plos.org